# HIF-1 stabilization in T cells hampers the control of *Mycobacterium tuberculosis* infection

Ruining Liu [1], Victoria Muliadi [1], Wenjun Mou[1,4], Hanxiong Li [1], Juan Yuan[2], Johan Holmberg[2], Benedict J. Chambers [3], Nadeem Ullah [1], Jakob Wurth[1,5], Mohammad Alzrigat [1], Susanne Schlisio [1], Berit Carow[1,6], Lars Gunnar Larsson[1] & Martin E. Rottenberg [1] ✉

The hypoxia-inducible factors (HIFs) regulate the main transcriptional pathway of response to hypoxia in T cells and are negatively regulated by von Hippel-Lindau factor (VHL). But the role of HIFs in the regulation of CD4 T cell responses during infection with *M. tuberculosis* isn't well understood. Here we show that mice lacking VHL in T cells (*Vhl cKO*) are highly susceptible to infection with *M. tuberculosis*, which is associated with a low accumulation of mycobacteria-specific T cells in the lungs that display reduced proliferation, altered differentiation and enhanced expression of inhibitory receptors. In contrast, HIF-1 deficiency in T cells is redundant for *M. tuberculosis* control. *Vhl cKO* mice also show reduced responses to vaccination. Further, VHL promotes proper MYC-activation, cell-growth responses, DNA synthesis, proliferation and survival of CD4 T cells after TCR activation. The VHL-deficient T cell responses are rescued by the loss of HIF-1α, indicating that the increased susceptibility to *M. tuberculosis* infection and the impaired responses of *Vhl*-deficient T cells are HIF-1-dependent.

Tuberculosis (TB), caused by infection with *Mycobacterium tuberculosis*, remains a leading public health problem. While most individuals infected with *M. tuberculosis* remain asymptomatic, in 2018 10 million individuals fell ill with this disease and 1.5 million died[1]. Why *M. tuberculosis* occurs only in some individuals is only partially understood. The risk of developing TB increases during HIV and *M. tuberculosis* co-infection suggesting that impairment of CD4 T cell-mediated immune responses reactivates the asymptomatic infection[2]. The central role for CD4 T cells in protective immunity against TB is also evidenced by the extreme susceptibility of animals lacking CD4 T cells[3]. $T_H1$ immune responses that involve cellular effector mechanisms such as IFN-γ mediated macrophage activation are utilized by the host to counteract mycobacterial infections[4–7].

Bacille Calmette–Guérin (BCG) the only TB vaccine shows a variable efficiency against pulmonary TB in adults[8], and a better vaccine against *M. tuberculosis* is urgently needed. Correlates of protective immune responses to *M. tuberculosis* infection in humans remain undefined. Understanding the detailed molecular regulation of T cell functions in the lung is paramount for building a strategy for vaccine induced protection against TB.

Survival, activation, and effector function of T cells is fundamentally linked to cellular metabolic programming[9]. A number of transcription factors have been implicated in the regulation of T cell metabolism and the T cell responses following activation[10]. Hypoxia is a hallmark trait of inflamed tissues. The hypoxic stress response is largely governed by hypoxia-inducible factors HIF-1α and HIF-2α, that

[1]Department of Microbiology, Tumor and Cell Biology, Karolinska Institutet, Stockholm, Sweden. [2]Department of Cell and Molecular Biology, Karolinska Institutet, Stockholm, Sweden. [3]Center for Infectious Medicine, Department of Medicine Huddinge, Karolinska Institutet, Stockholm, Sweden. [4]Present address: Capital Children's Hospital, Beijing, China. [5]Present address: University Children's Hospital Zurich, Zurich, Switzerland. [6]Present address: Novovax AB, Uppsala, Sweden. ✉e-mail: martin.rottenberg@ki.se

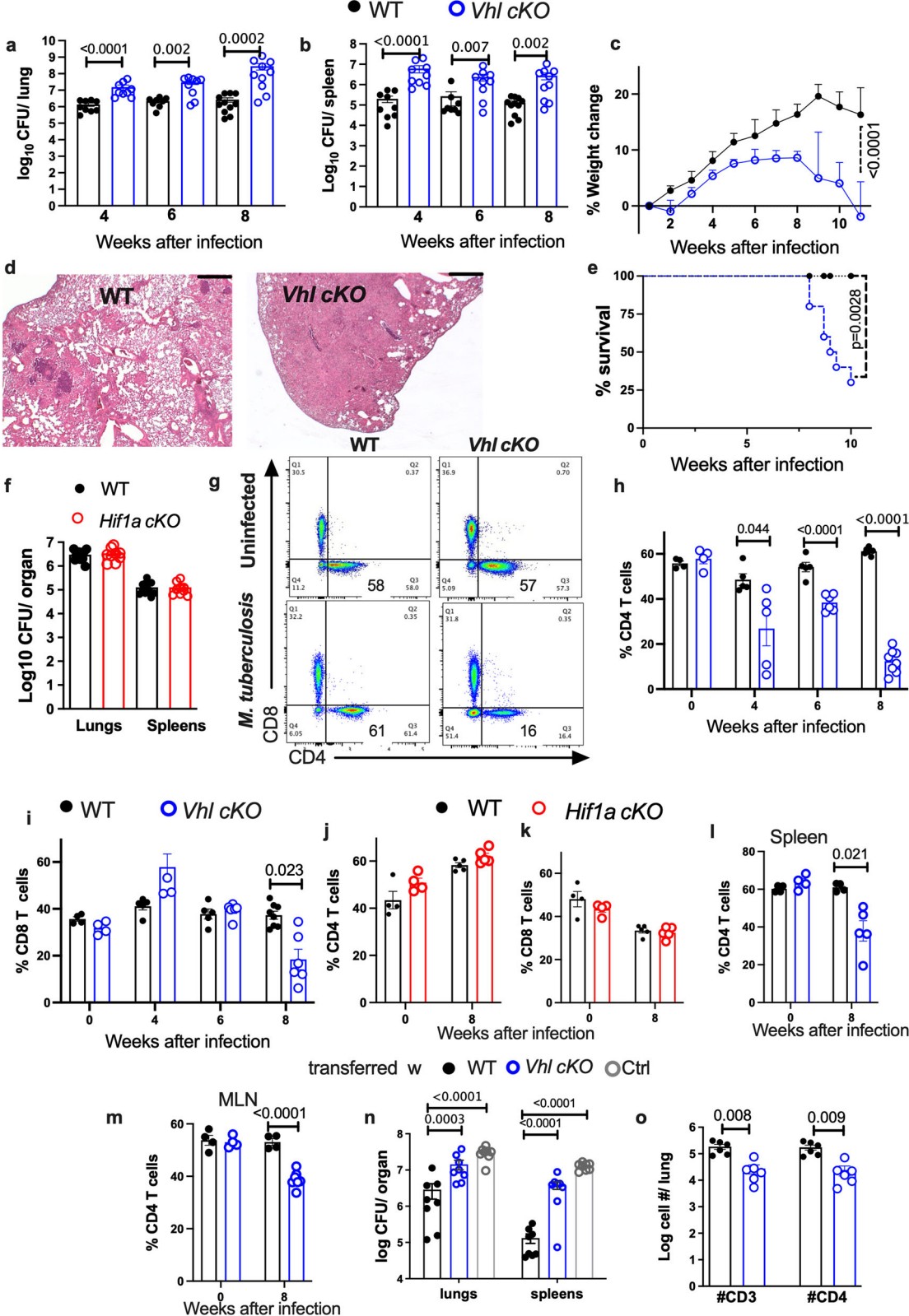

heterodimerize with HIF-1β and serve as the central sensor of oxygen tension in all cells. HIFs are highly relevant to the proper function of different immune cell populations including T cells[11]. In normal oxygen tensions, the HIF-α proteins are hydroxylated by prolyl hydroxylase domain proteins (PHD or EGLN proteins). These modifications are recognized by the von Hippel Lindau tumor suppressor protein (VHL),

an E3 ubiquitin ligase that drives the degradation of the HIF-α subunits[12–14]. HIF-1 and HIF-2 drive a metabolic reprogramming through the upregulation of glycolytic metabolism and the suppression of oxygen consumption by mitochondria[15]. In T cells, signals mediated by T cell antigen receptors (TCRs), cytokines and innate immune receptors have been shown to activate HIF-pathways even under normal

**Fig. 1 | VHL expression in T cells is critical for the control of *M. tuberculosis* infection in mice. a**, **b** *Vhl cKO* and WT mice were infected via aerosol with 250 *M. tuberculosis* and sacrificed at indicated time points after infection. The $\log_{10}$ CFU in the lung (**a**) and spleen (**b**) of individual mice ($n$ WT = 10, 9, 12; $n$ *Vhl cKO* = 9, 9, 10 mice at 4, 6 and 8 weeks post infection (w.p.i.) respectively) are represented. **c** The weight change with respect to the uninfected group mean (day 0) during infection with *M. tuberculosis* of *Vhl cKO* and WT mice ($n = 11$ per group) is depicted. **d** Representative micrographs from hematoxylin-eosin-stained paraffin lung sections from *Vhl cKO* and WT mice 8 w.p.i. with *M. tuberculosis* (bar: 400 μm). **e** The cumulative mortality of *Vhl cKO* and WT mice ($n = 9$ per group) after *M. tuberculosis* infection is depicted. **f** The $\log_{10}$ CFU in the lungs and spleens of *Hif1a cKO* ($n = 11$) and WT ($n = 9$) mice 8 w.p.i. with *M. tuberculosis* are shown. **g–i** The representative dot plots of CD4 and CD8 (gated on T cells) in the lungs of *Vhl cKO* and WT mice before and 8 w.p.i. with *M. tuberculosis* (**g**), and the frequency of CD4 (**h**) and CD8 (**i**) T cells in the lungs (WT $n = 4, 4, 6, 8$ and *Vhl cKO* $n = 4, 5, 6$ and 8 mice at 0, 4, 6 and 8 w.p.i. respectively). **j**, **k** The frequency of CD4 and CD8 T cells in the lungs of *Hif1a cKO* and WT mice before ($n = 4$ per group) and 8 w.p.i. ($n = 5$ per group) are depicted. **l**, **m** The frequency of CD4 in spleens (**l**) and mediastinal lymph nodes (**m**) before and 8 w.p.i. with *M. tuberculosis* are shown (*Vhl cKO* $n = 4, 7$ and WT $n = 4, 4$ mice at 0 and 8 w.p.i., respectively). $Rag2^{-/-}$ mice were administered i.v. with either $2.10^6$ *Vhl cKO* or WT CD4 T cells ($n = 6$) or left untreated 3 days after *M. tuberculosis* infection. The $\log_{10}$ CFU in the lungs and spleens (**n**) ($n = 8$ animals per group) and number of CD4 T cells (**o**) in lungs ($n = 6$ mice per group) 4 w.p.i. Each symbol represents one mouse, and the data are presented as the mean ± s.e.m. The $p$-values were calculated using either a two-tailed unpaired $t$ test with Welch's correction for no homoscedasticity and FDR approach for multiple comparison (**a**, **b**, **f**, **h–m**, **o**), a one-way ANOVA with Welch's correction (**n**), a 2-way ANOVA with Sidak's multiple comparison test (**c**), or a $\chi^2$ test (**e**). Source data are provided as a Source Data file.

oxygen tension[11]. Constitutive HIF activity achieved by conditional deletion of *Vhl* or *Phd*(s) had been shown to modulate the effector functions, the differentiation, and the expression of inhibitory receptor in CD4 and CD8 T cells, improving the control of tumors and viral infections[16–18].

Here, we studied the role of HIFs in the regulation of T cells during *M. tuberculosis* infection. In contrast to previous observation that VHL deficiency enhances effector responses of T cells, mice with VHL-loss in T cells (*Vhl cKO*) showed a dramatically increased susceptibility to infection with *M. tuberculosis* by failing to accumulate *M. tuberculosis*-specific CD4 and CD8 T cells in the lungs. Mutant T cells showed reduced levels of differentiation markers, increased expression of inhibitory receptors and low IFN-γ secretion. In contrast HIF-1-deficiency in T cells was redundant for protection against *M. tuberculosis* infection. We showed that VHL promoted CD4 T cell activation after T-cell receptor (TCR) stimulation. We demonstrated that the impaired TCR-mediated responses, the increased susceptibility to *M. tuberculosis* and the defect in mounting responses to immunization of *Vhl cKO* CD4 T cells are due to HIF-1 stabilization.

## Results

### VHL, but not HIF-1 expression in T cells is critical for the control of *M. tuberculosis* infection in mice

We studied the role of VHL and HIF-1 in T cells in the outcome of infection with 250 *M. tuberculosis* using *Vhl^{fl/fl} cd4 cre* (*Vhl cKO*) and *Hif1a^{fl/fl}dlck cre* (*Hif1a cKO*) mice. Lungs and spleens from *Vhl cKO* mice at 4, 6 and 8 weeks after aerosol infection all showed increased levels of *M. tuberculosis* as compared to those from *Vhl^{fl/fl}* controls (WT) (Fig. 1a, b). *Vhl cKO* mice lost weight as compared to *WT* controls (Fig. 1c). The lungs from *M. tuberculosis*-infected *Vhl cKO* mice showed lesions engaging a larger area of the lung as compared to those from controls (Fig. 1d). *Vhl cKO* mice also showed decreased cumulative survival after *M. tuberculosis* infection (Fig. 1e). The titers of *M. tuberculosis* in lungs and spleens from *Hif1a cKO* and *Hif1a^{fl/fl}* littermates (WT) were similar (Fig. 1f). The lesion severity in lungs from *Hif1a cKO* and WT mice was similar (Supplementary Fig. 1a), and no mortality was observed in these groups.

Next, we evaluated the impact of the HIF-1 and VHL-deficiency on the numbers, frequencies and phenotype of pulmonary T cells during *M. tuberculosis* infection. We found that the frequencies and numbers of T cells in lungs from *Hif1a cKO*, *Vhl cKO* and WT mice before and after *M. tuberculosis* infection were similar (Supplementary Fig. 1b–d). However, lungs from *M. tuberculosis*-infected *Vhl cKO* mice showed a reduced frequency of CD4 T cells at 4, 6 and 8 weeks after infection when compared to WT mice (Fig. 1g, h). The frequency of lung CD8 T cells at 8, but not at 4 and 6 weeks after infection, was decreased in *Vhl cKO* mice (Fig. 1g, i). Instead, the frequency and numbers of *Vhl cKO* CD4-CD8- (DN) T cells, consisting of a majority of γδT cells was higher in *Vhl cKO* than in WT controls (Supplementary Fig. 1e–h).

The frequencies of CD4, CD8 and DN T cells in lungs from *Hif1a cKO* and WT mice before or after infection with *M. tuberculosis* were instead similar (Fig. 1j, k and Supplementary Fig. 1i).

*Vhl cKO* mice showed low levels of CD4 T cells in spleens and mediastinal lymph nodes (MLN) at 8 weeks after infection (Fig. 1l, m). Of importance, the frequencies of CD4 and CD8 T cells in lungs, spleens and MLN from uninfected *Vhl cKO* and WT mice was similar (Fig. 1h, i, l, m). The total cell numbers in the MLNs in WT and *Vhl cKO* mice were similar, whereas MLN CD4 T cells numbers were reduced in *Vhl cKO M. tuberculosis* infected mice when compared to controls (Supplementary Fig. 1j, k).

CD4 T cells are critical for the control of *M. tuberculosis* infection[3,19]. To address whether VHL-deficiency in CD4 T cells hampers their protective capacity against *M. tuberculosis*, naïve *Vhl cKO* or WT CD4 T cells were transferred to $Rag2^{-/-}$ mice 3 days after infection with *M. tuberculosis*. The bacterial load in lungs and spleens from mice 4 weeks after transfer with WT CD4 T cells was lower than in those from non-transferred infected controls (Fig. 1n). $Rag2^{-/-}$ mice transferred with *Vhl cKO* CD4 T cells showed increased bacterial loads in lungs and spleens and lower number of CD4 T cells in the lung when comparing to mice transferred with WT CD4 T cells (Fig. 1n, o and Supplementary Fig. 1l, m). Thus, transfer of *Vhl cKO* CD4 T cells do not confer protection against *M. tuberculosis* infection.

### Low levels of mycobacteria-specific T cells in lungs of *Vhl cKO* infected mice

The levels of *M. tuberculosis*-specific CD4 and CD8 T cells in the lungs from *Vhl cKO*, *Hif1a cKO* and WT mice were then compared. The frequency of ESAT6_{1-20}-tetramer binding CD4 T cells in the lungs of *Vhl cKO* after infection with *M. tuberculosis* was reduced when compared to that of WT mice (Fig. 2a, b). Instead, the frequencies of ESAT6-tetramer binding cells in lungs from *M. tuberculosis*-infected *Hif1a cKO* and WT mice were similar (Fig. 2c).

*M. tuberculosis* TB10.4_{4-11} tetramer binding CD8 T cells were reduced in lungs from *Vhl cKO* infected mice (Fig. 2d, e), while levels of TB10.4 tetramer binding CD8 T cells in the lungs from *M. tuberculosis*-infected *Hif1a cKO* and WT mice were similar (Fig. 2f). In line with this observation, the fractions of IFN-γ secreting CD4 and CD8 T cells from the lungs from *M. tuberculosis*-infected *Vhl cKO* mice, stimulated respectively with ESAT-6_{1-20} or TB10.4 _{4-11} peptides were low when comparing to those of WT cells (Fig. 2g–i).

T cells proliferate in the lung during the infection with *M. tuberculosis*[20]. We observed during *M. tuberculosis* infection, that lung *Vhl cKO* CD4 T cells proliferated less when compared to WT controls as measured by the expression of Ki-67 proliferation marker (Fig. 2j). The frequency of Ki-67 expressing TB10.4 tetramer-binding CD8 T cells in lungs from *M. tuberculosis*-infected *Vhl cKO* was also lower than those the from the WT mice (Fig. 2k). The frequency of apoptotic, Annexin V + CD4 and CD8 T cells was high in lungs from *M. tuberculosis*-infected *Vhl cKO* mice (Fig. 2l).

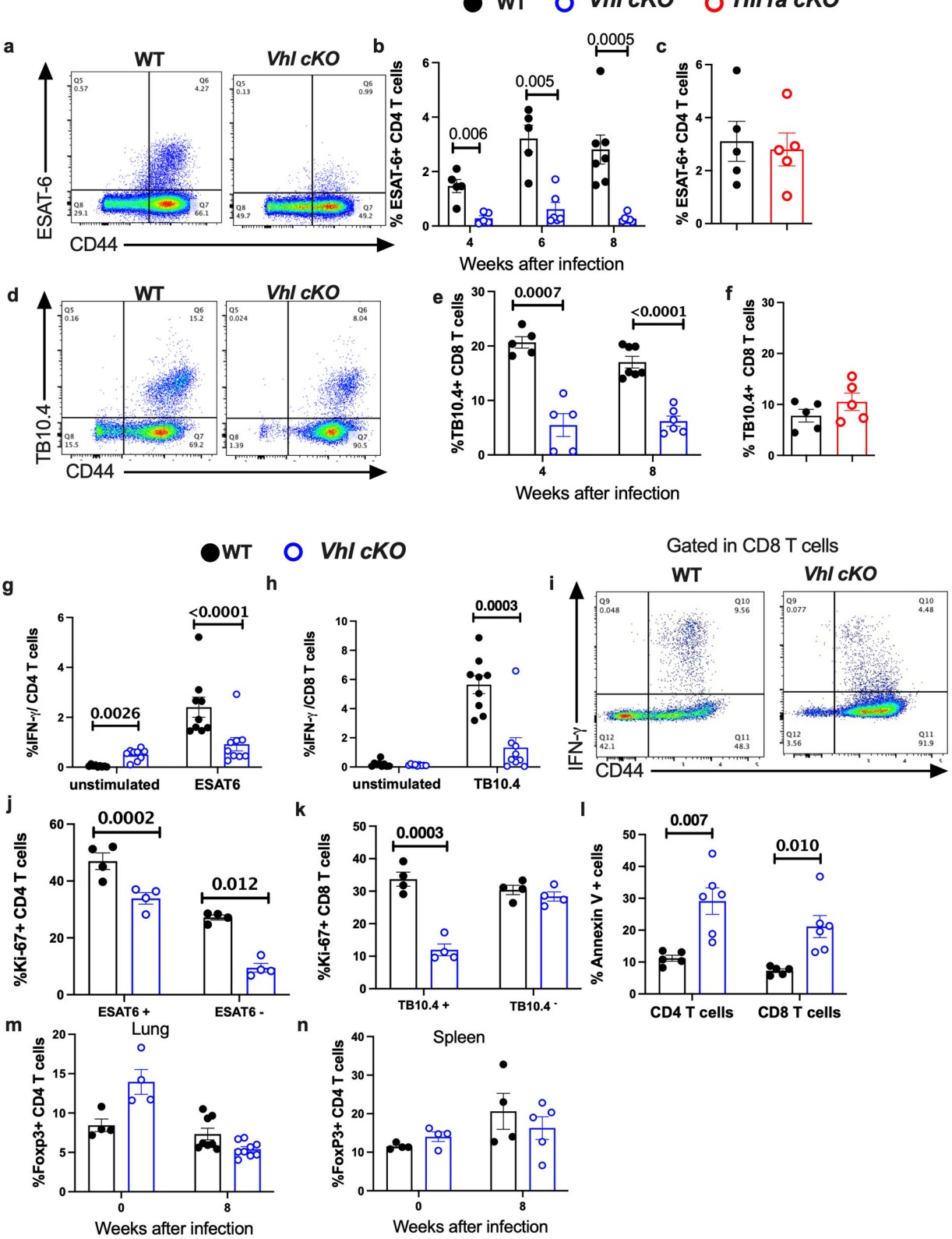

We investigated if increased regulatory T cells (Tregs) could account for the impaired proliferation of CD4 and CD8 T cells of *Vhl cKO* mice and for the enhanced susceptibility of *Vhl cKO* mice to *M. tuberculosis* infection. We found that the frequency of CD25 + Foxp3 + CD4 T cells in lungs and spleens from *M. tuberculosis*-infected was similar (Fig. 2m, n).

**Increased expression of inhibitory receptors in T cells from *Vhl cKO* mice infected with *M. tuberculosis***

Then, the impact of VHL and HIF-1 expression in T cell differentiation in the lung of *M. tuberculosis* infected mice was studied. We observed a preferential increase of effector memory T cells ($T_{EM}$) in lungs of WT mice infected with *M. tuberculosis*

**Fig. 2 | Reduced accumulation of mycobacteria-specific T cells in the lungs of *M. tuberculosis* infected *Vhl cKO* mice. a** Representative dot plots of tetramer ESAT-6-binding CD4 T cells in the lungs from *Vhl cKO* and WT mice 8 weeks post infection (w.p.i.) with *M. tuberculosis* are depicted. **b** The percentage of tetramer ESAT-6-binding CD4 T cells in the lungs of *Vhl cKO* and WT mice (WT $n = 5, 5, 7$ and *Vhl cKO* $n = 5, 7, 7$ mice at 4, 6 and 8 w.p.i.) are displayed. **c** Percentage of tetramer ESAT-6 + CD4 T cells in the lungs of *Hif1a cKO* ($n = 5$) and WT ($n = 5$) mice 8 w.p.i. (**d, e**) Dot plots (**d**) and the frequency of tetramer TB10.4-binding CD8 T cells (**e**) in the lungs from *Vhl cKO* ($n = 5, 6$) and WT mice ($n = 5, 7$ animals at 4 and 8 w.p.i., respectively) are depicted. **f** The frequency of tetramer TB10.4-binding lung *Hif1a cKO* ($n = 5$) and WT ($n = 5$ mice) CD8 T cells at 8 w.p.i. is depicted. **g, h** The frequencies of IFN-γ secreting CD4 (**g**) and CD8 (**h**) T cells from *Vhl cKO* ($n = 9$) and WT ($n = 9$) mice at 8 wp.i. stimulated with either ESAT-6$_{1-15}$ or TB10.4$_{4-11}$ peptides or left untreated are shown. **i** Dot plots showing the IFN-γ secretion by lung CD8 T cells from *Vhl cKO* and WT mice 8 weeks after *M. tuberculosis* infection. **j, k** The frequency of Ki-67 + , tetramer-binding and total CD4 (**j**) and CD8 (**k**) T cells from lung of *Vhl cKO* ($n = 5$) and WT mice ($n = 5$) 8 w.p.i. **l** The fraction of lung Annexin V + CD4 and CD8 T cells from *Vhl cKO* ($n = 5$) and WT ($n = 6$) mice at 8 weeks after *M. tuberculosis* infection is shown. **m, n** The percentage of Foxp3+ CD4 T cells in the lungs (**m**) and spleens (**n**) of *Vhl cKO* and WT mice at 0 and 8 w.p.i. (*Vhl cKO* $n = 5$, WT $n = 4$) is depicted (lungs 0 w.p.i. WT $n = 4$, *Vhl cKO* $n = 4$; 8 w.p.i. WT $n = 5$ *Vhl cKO* $n = 8$; spleens 0 w.p.i. $n = 4$, *Vhl cKO* $n = 4$; 8 w.p.i WT $n = 4$, *Vhl cKO* $n = 5$). Each symbol represents one mouse, and the data are presented as the mean ± s.e.m. The *p* values were calculated using a two-tailed unpaired *t* test with Welch's correction and FDR adjustment for multiple comparisons. Source data are provided as a Source Data file.

(Supplementary Fig. 2a–d). Lower frequencies of CD4 but not CD8 $T_{EM}$ in lungs from *Vhl cKO* and WT *M. tuberculosis*-infected mice were found (Supplementary Fig. 2a, b). *Vhl cKO* mice showed a high frequency of CD44 + CD62L + central memory CD8 T cells ($T_{CM}$) in lungs before and during infection, while CD4 $T_{CM}$ were only increased in *Vhl cKO* at 8 weeks after infection (Fig. 3a–c). The frequency of TB10.4 tetramer-binding within CD8 $T_{CM}$ cells in lungs from WT and *Vhl cKO M. tuberculosis*-infected mice was lower than that of TB10.4 + CD8 $T_{EM}$ cells (Fig. 3d). The percentage of TB10.4 + CD8 $T_{CM}$ in the lungs of *M. tuberculosis* infected *Vhl cKO* and WT mice was similar. Instead, the fraction of tetramer TB10.4-binding $T_{EM}$ cells was lower in the lungs of *Vhl cKO M. tuberculosis*-infected mice as compared to WT controls (Fig. 3d). Similar percentages of CD4 and CD8 $T_{CM}$ and $T_{EM}$ from *Hif1a cKO* and WT mice infected with *M. tuberculosis* were determined (Fig. 3e, f and Supplementary Fig. 2c, d).

PD-1 + T cells display proliferative and protective capacity and residence in the lung parenchyma after *M. tuberculosis* infection, while KLRG1 is expressed in short-lived terminally differentiated effector cells[21–23]. We found that the cell frequency and the expression levels of KLRG1 were diminished in *Vhl cKO* CD4 and CD8 T cells as compared to WT controls (Fig. 3g–i, Supplementary Fig. 2e, f). While the frequency of PD-1 + CD4 and CD8 T cells in lungs from *M. tuberculosis*-infected *Vhl cKO* and WT mice was similar, the expression levels of PD-1 were higher in pulmonary *Vhl cKO* CD4 and CD8 T cells from *M. tuberculosis*-infected mice (Fig. 3j–m and Supplementary Fig. 2g, h).

The expression of the inhibitory receptor CTLA-4 increased in CD4 (but not CD8) T cells after *M. tuberculosis* infection. The percentage of lung CD4 and CD8 T cells expressing CTLA-4 was higher in *Vhl cKO* infected mice when comparing with WT controls (Fig. 3n, o).

The T cell chemokine receptor expression pattern determines the ability to migrate into the lungs to protect against *M. tuberculosis* infection. Less-differentiated T cells that express CXCR3 can migrate into the lungs and suppress the growth of *M. tuberculosis*[21,24]. In contrast, T cells that express high levels of CX3CR1 poorly migrate out of the blood vessels[25,26]. We found reduced frequencies of both CXCR3 + and CX3CR1 + CD4 T cells from lungs of *Vhl cKO* mice 8 weeks after infection with *M. tuberculosis* when compared to those of WT controls (Fig. 3p–r), while the fractions of CXCR3 + and CX3CR1 + CD8 T cells in lungs from *Vhl cKO* and WT *M. tuberculosis* infected mice were similar (Fig. 3q, r). In contrast, the frequency of tetramer TB10.4 + CXCR3 + CD8 T cells in *Vhl cKO* was lower than WT controls (Supplementary Fig. 2i).

To determine if vascular localization of CD4 and CD8 T cells in *M. tuberculosis*-infected *Vhl cKO* mice was impaired, intravascular labeling with anti-CD45.2 mABs was performed. We found similar frequencies of extravascular (CD45.2 neg) CD4 and CD8 T cells in the lungs of *M. tuberculosis*-infected WT and *Vhl cKO* mice (Fig. 3s), although the frequency of *Vhl cKO* extravascular tetramer TB10.4 + T cells was low (Fig. 3t). More than 80% of CXCR3 + or PD-1 + T cells located in the parenchyma (Supplementary Fig. 2j, k, m). Instead, 10% of CX3CR1 + and 20% of KLRG1 + CD4 or CD8 T cells were in the lung parenchyma of WT mice (Supplementary Fig. 2l, n, o). The frequencies of PD-1 + , CXCR3 + , CX3CR1 + or KLRG1 + T cells that locate in the lung parenchyma of WT and *Vhl cKO* mice were similar (Supplementary Fig. 2k–m, o).

Altogether, during the *M. tuberculosis* infection of *Vhl cKO* mice, lung CD4 T cells showed decreased KLRG1, CXCR3, CX3CR1 and increased PD-1 and CTLA-4 levels. Some of these deviations were also observed in mutant CD8 T cells.

## VHL promotes proliferative and effector gene expression programs in CD4 T cells from *M. tuberculosis*-infected mice

We then compared the global mRNA expression in sorted lung CD4 T cells from *Vhl cKO* and WT mice 8 weeks after infection with *M. tuberculosis*. We found that 3524 genes were upregulated and 3086 downregulated in *Vhl* deficient vs WT CD4 T cells (Fig. 4a and Supplementary Fig. 3a). Heatmaps and unsupervised PCA analysis showed a differential clustering of WT and *Vhl cKO* CD4 T cells (Fig. 4b, c). Genes involved in inhibition of T cells responses, glycolysis, and prolyl hydroxylases were among the 25 utmost differentially expressed in *Vhl cKO* CD4 T cells (Fig. 4a). The gene hallmark analysis showed that hypoxia, angiogenesis, glycolysis pathways were increased in *Vhl cKO* CD4 T cells (Fig. 4d), while DNA replication and metabolism, oxidative phosphorylation, MYC, E2F pathways were lower (Fig. 4e). Within the HIF-1 pathway, glycolytic, oxygen sensing, angiogenesis related genes genes and the glucose receptor *Slc2a1* were increased in *Vhl cKO* CD4 T cells (Fig. 4f). The WT CD4 T cells were enriched in genes involved in lymphocyte differentiation and chemokine responses (Supplementary Fig. 3b, d), while several transcripts involved T cell exhaustion or disfunction were increased in *Vhl cKO* CD4 T cells (Fig. 4g).

*Mxi1* transcripts encoding the MAX-interactor 1, a HIF-target and negative regulator of MYC expression and activity[27] was upregulated in *Vhl cKO* CD4 T cells. In line with a negative regulation of MYC activity, the cell cycle inhibitors *Cdkn1a* (p21) and *Cdkn1b* (p27) which are repressed by MYC[28] were increased and the cyclin dependent kinase *Cdk1* transcripts were diminished in *Vhl cKO* CD4 T cells (Fig. 4h). MYC and HIF-1 regulate the expression of numerous genes involved in mitochondrial biogenesis, including genes that control transcription and translation such as the mitochondrial DNA-directed RNA polymerase (*Polrmt*), and the transcription factor A mitochondrial (*Tfam*)[29], which were found to be reduced in *Vhl cKO* CD4 T cells from *M. tuberculosis*-infected mice (Fig. 4h). Moreover, the HIF-1 target *Bnip3* and *Bnip3l*, coding for molecules that promote autophagy and mitophagy were increased in *Vhl cKO* CD4 T cells (Fig. 4h and Supplementary Fig. 3c, e).

Since metabolic pathways were widely regulated in *Vhl cKO* CD4 T cells during *M. tuberculosis* infection, we measured the mitochondrial bioenergetic profiles of lung T cells during infection with *M. tuberculosis*. We found that the mitochondrial mass was reduced in *Vhl*

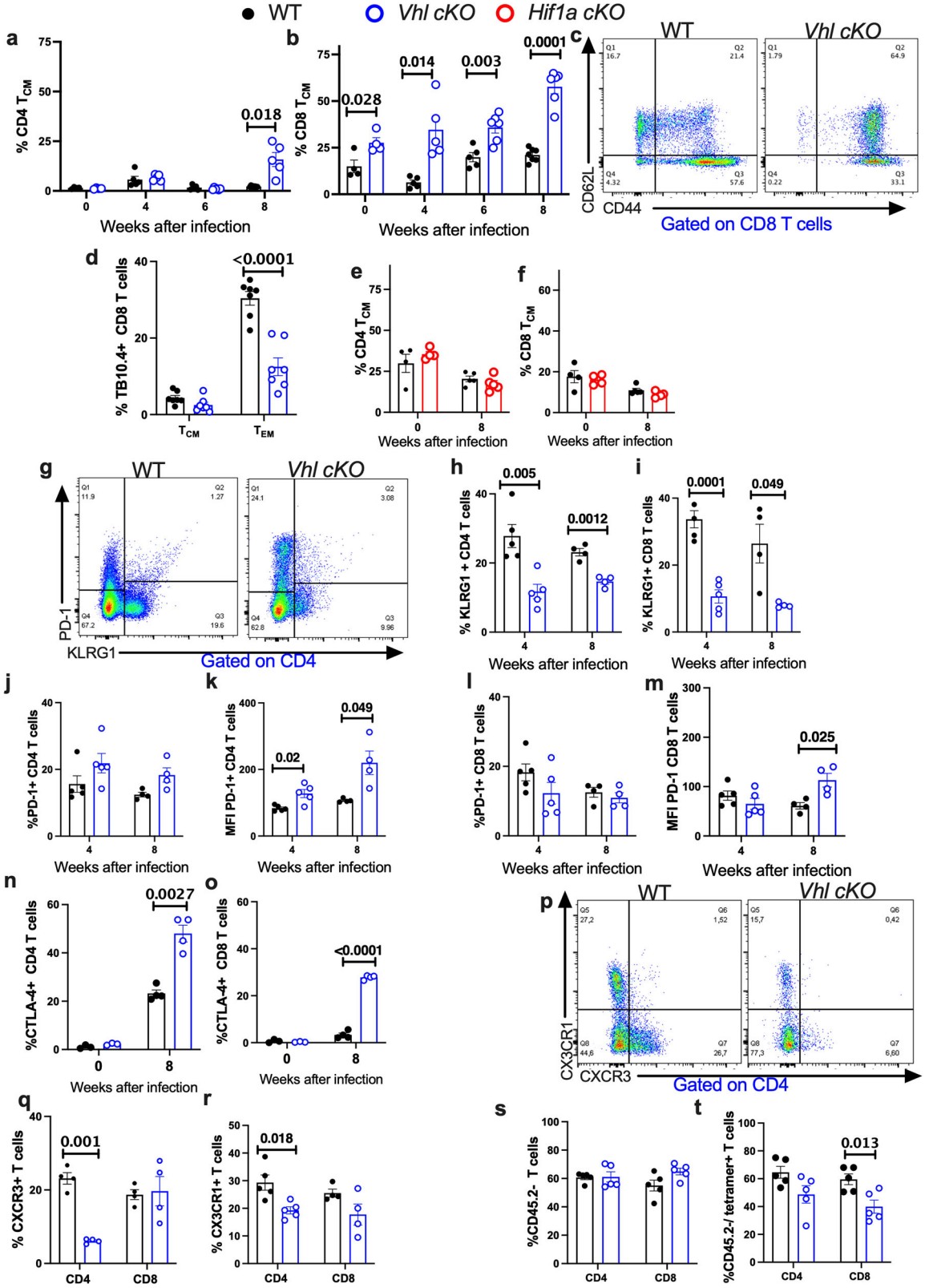

*cKO* CD4 T cells from infected mice (Fig. 4i). Moreover, levels of mitochondrial ROS were also decreased in *Vhl cKO* CD4 and CD8 T cells from infected mice as compared to controls (Fig. 4j).

## VHL expression is required for TCR activation of CD4 T cells

Results obtained above suggested that VHL expression in CD4 T cells was critical for DNA synthesis, cell cycle progression and MYC activity.

To further explore the mechanisms involved, the responses of naïve *Vhl cKO* CD4 T cells to T cell receptor stimulation in vitro were studied. The CD4 T cell activation increased the levels of HIF-1α in both WT and *Vhl*-mutant CD4 T cells. Sorted *Vhl cKO* CD4 T cells showed, as expected, higher levels of HIF-1α when compared to WT controls before and after TCR stimulation (Fig. 5a). In line with this, the levels of the HIF-1 regulated transcripts *Vegfa* and *Ldha* were increased in *Vhl*

**Fig. 3 | Decreased expression of differentiation markers and enhanced levels of inhibitory receptors in *Vhl cKO* T cells during *M. tuberculosis* infection. a–c** The frequency of lung CD4 (**a**) and CD8 (**b**) $T_{CM}$ (CD44 + CD62L + ) from *Vhl cKO* ($n = 4$, 5, 6, 7) and WT ( *Vhl cKO*; WT $n = 4$, 5, 5 and 7 mice at 0, 4, 6, and 8 weeks after *M. tuberculosis* infection are shown. **c** Dot plot of CD44 and CD62L in lung CD8 T cells from *Vhl cKO* and WT mice 8 weeks after infection. **d** The frequencies of tetramer TB10.4-binding pulmonary CD8 $T_{CM}$ and $T_{EM}$ cells from *Vhl cKO* ($n = 6$) and WT ($n = 7$) mice 8 weeks after *M. tuberculosis* infection are depicted. **e, f** The fraction of pulmonary CD4 (**e**) and CD8 (**f**) $T_{CM}$ from *Hif1a cKO* and WT mice at 0 ($n = 4$ per group) and 8 ($n = 5$ per group) weeks after *M. tuberculosis* infection are depicted. **g–m** Dot plots (**g**) and frequency of KLRG1 + and PD-1 + CD44 + CD4 (**h, j**) and CD8 (**i, l**) T cells in the lungs of *Vhl cKO* and WT mice at 4 and 8 weeks after *M. tuberculosis* infection. The MFI of PD-1 in CD4 (**k**) and CD8 T (**m**) cells in the lungs of *Vhl*

*cKO* and WT mice is also shown (**h–m**; $n = 5$ and 4 mice per group at 4 and 8 weeks after infection). **n, o** The fractions of CTLA-4 + CD4 (**n**) and CD8 (**o**) T cells in the lungs of mice before ($n = 3$) and 8 weeks ($n = 4$ mice per group) after infection are depicted. **p–r** Dot plots (**p**) and frequencies of CXCR3+ (**q**) and CX3CR1+ (**r**) CD4 and CD8 T cells in the lungs of *Vhl cKO* and WT mice ($n = 4$ per group) 7 weeks after *M. tuberculosis* infection. **s, t** The frequency of lung parenchymal (i.v. CD45.2 negative) total (**s**) and tetramer binding (**t**) CD4 and CD8 T cells in the lungs of WT and *Vhl cKO* mice ($n = 5$ per group) 8 weeks after infection with *M. tuberculosis* are shown. Each symbol represents one mouse, and the data are presented as the mean ± s.e.m. The *p* values were calculated using a two-tailed unpaired *t* test with Welch's correction and FDR approach for multiple comparison. Source data are provided as a Source Data file.

*cKO* CD4 T cells before and after TCR stimulation (Supplementary Fig. 4a, b).

The in vitro cell cycle profile of mutant and control CD4 T cells was compared. The percentage of *Vhl cKO* CD4 T cells in S phase was markedly reduced when compared to that of WT controls already 16 h after anti-CD3/CD28 stimulation (Fig. 5b, d). A large proportion of *Vhl cKO* CD4 T cells were in the G2/M phase compared to controls (Fig. 5c, d). Since the oncoprotein/transcription factor MYC supports cell cycle progression, the protein expression of MYC in *Vhl cKO* and WT CD4 T cells was measured. MYC was rapidly induced in WT and *Vhl cKO* CD4 T cells after TCR stimulation, but the level of MYC in *Vhl cKO* CD4 T cells after TCR-stimulation was attenuated (Fig. 5e). The *Myc* mRNA levels in WT and *Vhl cKO* CD4 T cells before or after TCR stimulation were similar, suggesting that the reduction in MYC protein level in *Vhl cKO* CD4 T cells is regulated at the post-transcriptional level (Fig. 5f). The anti-CD3/CD28 triggering of WT CD4 T cells increased the expression of CD71, the transferrin receptor, a downstream target of MYC. The expression level and the frequency of CD71 expressing *Vhl cKO* CD4 T cells was low when compared with WT controls (Fig. 5g, h). The accumulation of *Cd71* mRNA was also reduced in anti-CD3/CD28-stimulated *Vhl cKO* CD4 T cells when compared to controls (Fig. 5i). The titers of *Mxi1* mRNA, coding for a MYC-negative regulator, and of *Cdkn1a* mRNA was higher in *Vhl cKO* than in WT CD4 T cells (Supplementary Fig. 4c, d).

We also found a reduced level of expression of the TCR-β, CD3ε and CD4 receptors in *Vhl cKO* CD4 T cells compared to that in WT cells (Fig. 5j–m and Supplementary Fig. 4e, f), indicating that differences in the expression of diverse components of the TCR complex prior to stimulation can account for the diminished activation of *Vhl cKO* CD4 T cells.

## VHL promotes proliferation of CD4 T cells after TCR stimulation

MYC drives both cell growth and proliferation. We observed that the frequency of *Vhl cKO* CD4 T cells after anti-CD3/CD28 stimulation of splenocytes was low, resembling changes in T cell frequencies during infection in vivo (Fig. 6a–c). Instead, the frequencies of *Hif1a cKO* and WT T cell populations were similar (Fig. 6a–c). The proliferation of TCR-stimulated *Vhl cKO* CD4 T cells was impaired while that of CD8 T cells was reduced when compared to WT controls (Supplementary Fig. 5a–c). The proliferative responses of anti-CD3/CD28-stimulated naïve *Vhl cKO* CD4 T cells were abolished (Fig. 6d, e).

The growth response of *Vhl cKO* CD4 T cells after anti-CD3/ CD28 stimulation was markedly reduced when compared to WT controls (Fig. 6f–h and Supplementary Fig. 5d).

T cells express CD69 and CD44 early after TCR triggering. The expression density and the frequencies of CD69 + and CD44 + CD4 T cells increased from 24 to 48 h after anti-CD3/CD28 stimulation (Fig. 6i, j and Supplementary Fig. 5e–h). Despite the reduced growth and proliferative responses, the frequencies of CD69 and CD44 expressing *Vhl cKO* and WT CD4 T cells and their expression levels 24 h after anti CD3/CD28 stimulation were similar (Fig. 6i, j and

Supplementary Fig. 5e–h). Instead, 48 h after TCR stimulation the percentages of CD69 + and CD44 + *Vhl cKO* CD4 T cells were low compared to those of WT cells (Fig. 6i, j and Supplementary Fig. 5e, h). The intensity of CD44 on *Vhl cKO* CD4 T cells, but not that of CD69, 48 h after anti-CD3/CD28 stimulation was lower when compared to WT controls (Supplementary Fig. 5e–h).

IL-2 augments T cell proliferation and survival. The accumulation of IL-2 in supernatants of anti-CD3/CD28 stimulated *Vhl cKO* CD4 T cells cultures was impaired (Fig. 6k), in line with their deficient proliferative responses. The expression of CD25, the α-chain of the IL-2 receptor, was upregulated in naïve CD4 T cells after TCR activation (Fig. 6l, m). The expression levels of CD25 were also lower in *Vhl cKO* CD4 T cells than in WT controls (Fig. 6l, m). Thus, both IL-2 production and the expression of the high affinity IL-2 receptor were muted in *Vhl cKO* CD4 T cells after stimulation with anti-CD3/CD28. The addition of exogenous IL-2 did neither restore the proliferation, the CD44 expression nor resulted in increased cell size of TCR-stimulated *Vhl cKO* CD4 T cells (Supplementary Fig. 5i–k).

The frequency of Annexin V expressing TCR-triggered *Vhl cKO* CD4 T cells was high compared to WT controls (Fig. 6n and Supplementary Fig. 5l).

Homeostatic proliferation of T cells occurs when small numbers of T cells are adoptively transferred into immunodeficient hosts[30]. WT but not *Vhl cKO* CD4 T cells were detected in the blood and spleens of recipient *Rag2−/−* mice single transferred or co-transferred with $2.10^6$ WT or/and *Vhl cKO* CD4T cells (Fig. 6o-q and Supplementary Fig. 5m, n).

## HIF-1 mediates the susceptibility to *M. tuberculosis*-infection of *Vhl cKO* mice

To investigate whether HIF-1 overexpression mediates the increased *M. tuberculosis* susceptibility of *Vhl cKO* mice, *Hif1a[fl/fl] Vhl[fl/fl] cd4 cre (Vhl Hif1a dcKO)* mice were generated. *Vhl Hif1a dcKO* and WT mice showed similar bacterial levels in spleens and lungs 4 and 7 weeks after aerosol *M. tuberculosis* infection (Fig. 7a, b). The proportion and numbers of T cells and the frequencies of CD4 and CD8 T cells in lungs of *Vhl Hif1a dcKO* and WT mice at 4 and 7 weeks after *M. tuberculosis* infection were similar (Fig. 7c, d and Supplementary Fig. 6a–d). The percentage of ESAT-6 and TB10.4 tetramer-binding CD4 and CD8 T cells that were reduced in *Vhl cKO* infected compared to WT mice, were similar in WT and *Vhl Hif1a dcKO* mice (Fig. 7e, f). Lungs of *Vhl Hif1a dcKO* and WT mice also showed similar frequencies of central and effector memory CD4 and CD8T cells at 4 and 7 weeks after infection (Fig. 7g, h and Supplementary Fig. 6e–h). Further, the expression levels of PD-1, KLRG1, CXCR3 and CX3CR1 in CD4 T cells from WT and *Vhl Hif1a dcKO* mice at 4 and 7 weeks after *M. tuberculosis* infection were comparable (Fig. 7i–l and Supplementary Fig. 6i–l). The fraction of PD-1-, KLRG1-, CXCR3- and CX3CR1-expressing CD8 T cells from WT and *Vhl Hif1a dcKO* was also similar (Fig. 7m–p, Supplementary Fig. 6m–p). Altogether,

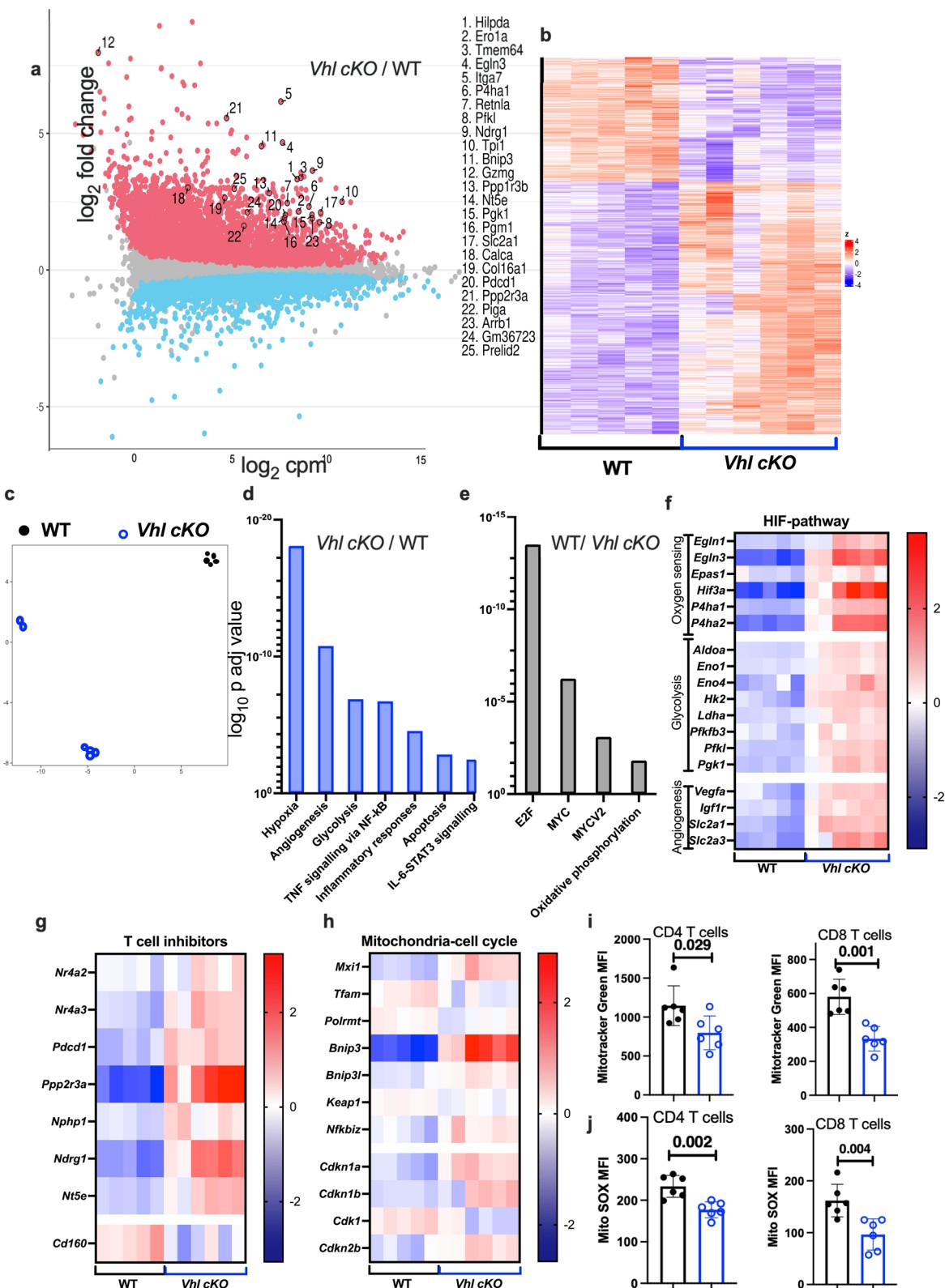

HIF-1 stabilization mediates the susceptibility of *Vhl cKO* mice during infection with *M. tuberculosis* and the distinct phenotype of T cells of *Vhl cKO* during the bacterial infection.

## HIF-1 stabilization in T cells impairs the responses to immunization with BCG

Whether VHL-deficiency in T cells also regulated the responses to immunization with the attenuated *M. bovis* BCG was studied. *Vhl cKO*,

*Vhl Hif1a dcKO* and WT mice were administered with BCG i.v. and the T cell phenotype in lungs and spleens analysed 3 weeks after immunization. The *Vhl cKO* mice showed lower percentage of CD4 T cells in lungs and spleens as compared to WT and *Vhl Hif1a dcKO* after BCG immunization, while the frequencies of the T cell populations in non-immunized mice were similar (Fig. 8a, b and Supplementary Fig. 7a). The frequency of CD8 T cells (but not of DN T cells) was increased in the lungs (but not spleens) of *Vhl cKO* mice as compared to WT and *Vhl*

**Fig. 4 | VHL expression in lung CD4 T cells from *M. tuberculosis*-infected mice controls proliferation and effector responses.** RNA seq was performed in *Vhl cKO* (n = 6) and WT (n = 5) CD4 T cells isolated from the lungs of *M. tuberculosis*-infected mice. **a** MA plot in which each point represents a gene. The plot illustrates the change of expression (y axis, $\log_2$ fold change) and the average transcript abundance over all samples (x axis, $\log_2$ counts per million). Colors indicate the significantly upregulated (in red) or down-regulated (in light blue) in *Vhl cKO* as compared to WT CD4 T cells. The top 25 most significant genes are indicated by their gene symbols. **b** In the heat map the $\log_2$ counts for each gene (row) is standardized to mean = 0, and the differences with the mean depicted. **c** Umap dimensionality reduction plots from non-supervised samples based on the normalized gene counts after filtering the low expressed genes. There are 2 outliers in the *Vhl cKO* CD4 T cell group, probably explained by their lower sample weight. **d**, **e** The $\log_{10}$ p value of the enrichment of hallmark gene sets in the transcriptome of *Vhl cKO* (**d**) and WT (**e**) CD4 T cells was compared using the MSigDB databases. **f**–**h** The heat maps of selected RNA-seq data showing: HIF-pathway-specific genes (**f**), genes involved in T cell exhaustion or dysfunction (**g**), and MYC-regulated transcripts (**h**). Data were normalized by subtracting the $\log_2$ transformed values to the mean $\log_2$ value for all samples for each gene. **i**, **j** Lung cell suspensions were labeled with mitochondrial-selective MitoTracker Green to assess mitochondrial mass and Mito Tracker Red to measure mitochondrial ROS. The MFI of Mito Tracker green, to assess mitochondrial mass (**i**), and Mito SOX Red, to assess mitochondrial ROS, (**j**) labeled live CD4 and CD8 T cells from the lung of *Vhl cKO* and WT mice (n = 6 per group) 8 weeks after infection with *M. tuberculosis* are shown. **i**, **j** Each symbol represents one mouse, and the data are presented as the mean ± s.e.m. **d**–**j** The p values were calculated using a two-tailed unpaired t test with Welch's correction (**k**, **m**) and FDR adjustment for multiple comparisons (**d**–**h**). Source data are provided as a Source Data file.

*Hif1a dcKO* mice (Fig. 8c and Supplementary Fig. 7b). Since ESAT-6 is not expressed by BCG, I-A$_b$ Ag85b$_{280-294}$ tetramers were used to quantify BCG-specific CD4 T cells in immunized mice. The frequency of tetramer Ag85b-binding CD4 T cells and of TB10.4-binding CD8 T cells was reduced in lungs and spleens of immunized *Vhl cKO* when compared with *Vhl Hif1a dcKO* and WT mice (Fig. 8d–g and Supplementary Fig. 7c, d). Further, the frequencies of CD4 and especially that of CD8 T$_{CM}$ in lungs and spleens from immunized *Vhl cKO* were higher (Fig. 8h, i and Supplementary Fig. 7g, h), whereas levels of CD4 and CD8 T$_{EM}$ were reduced when comparing to WT and *Vhl Hif1a dcKO* mice (Fig. 8j, k and Supplementary Fig. 7e, f). The frequency of CD8 T$_{CM}$ in the lungs and spleens of non-immunized *Vhl cKO* was higher than those of WT and *Vhl Hif1a dcKO* mice (Fig. 8i and Supplementary Fig. 7h).

CD8 T cells may exhibit a memory phenotype without overt immunization or infection. These cells, unlike true memory T cells that develop in response to foreign antigen, express only low levels of CD49d and are termed virtual memory T (T$_{VM}$) cells[31–33]. The CD44 + CD62L + CD8 T cells in spleens and lungs from non-immunized mice expressed low levels of CD49d suggesting these are T$_{VM}$ cells (Fig. 8l and Supplementary Fig. 7i). The frequency of CD49d expressing WT CD8T$_{CM}$ increased after immunization while levels in remained low in the *Vhl cKO* CD8T cells (Fig. 8l).

The fractions of KLRG1 + and CX3CR1 + CD4 and CD8 T cells in lungs and spleens from *Vhl cKO* mice were lower than those in WT and *Vhl Hif1a dcKO* mice (Fig. 8m–p and Supplementary Fig. 7j–m). Lower percentages of CXCR3 + CD4, but not CD8 T cells, were also recorded in the spleens (but not lungs) of *Vhl cKO* BCG immunized mice (Supplementary Fig. 7n, o). Different to infected mice, PD-1 levels were not increased in pulmonary or splenic *Vhl cKO* T cells as compared to those of WT or *Vhl Hif1a dcKO* mice (Supplementary Fig. 7p, q). However, the frequencies of CTLA-4 + CD4 and CD8 T cells from lungs of BCG-immunized *Vhl cKO* mice were higher than those measured in WT controls (Fig. 8q).

We found that the BCG CFU titers in the spleens and lungs from *Vhl cKO* were higher than those in WT mice, indicating that either clearance or control of dissemination of BCG was impaired in *Vhl cKO* mice. No overt signs of clinical disease were observed during the experiment (Supplementary Fig. 7r).

Altogether, *Vhl*-deficiency resulted in a HIF-1 dependent reduction of the pulmonary and splenic CD4 T cells and of specific T cell responses after BCG-immunization. As observed during infection the levels of the differentiation markers CX3CR1 and KLRG1 were reduced and the expression of CTLA-4 increased in T cells from immunized *Vhl cKO* mice.

### HIF-1 stabilization impairs CD4 T cell responses in vitro
We then studied whether HIF-1α overexpression accounts for the impaired TCR responses of *Vhl*-deficient CD4 T cells. We found that the growth response (Fig. 9a, b), the fraction of different cell cycle phases (Fig. 9c, d), the proliferation (Fig. 9e and Supplementary Fig. 8a) and

the CD44 expression levels (Supplementary Fig. 8b) in WT and *Vhl Hif1a dcKO* CD4 T cells at different times after TCR-activation were similar.

The transcriptome profile of WT, *Vhl cKO* and *Vhl Hif1a dcKO* CD4 T cells stimulated with anti-CD3/CD28 was then compared. The correlation analysis of the biological replicates (Fig. 9f), the heat map (Fig. 9g) and the principal component analysis (Supplementary Fig. 8c) showed the segregation of the transcriptome of unstimulated WT CD4 T cells in comparison to the other groups. Within the TCR-stimulated groups the gene expression of *Vhl cKO* CD4 T cells differed to that of WT and *Vhl Hif1a dcKO* CD4 T cells that showed important similarities between them (Fig. 9f–g). The *Vhl cKO* CD4 T cells expressed higher number of unique transcripts (499) and lower number of common genes as compared to either *Vhl Hif1a cKO* or WT CD4 T cells after anti-CD3/CD28 stimulation (Supplementary Fig. 8d). Eight hundred thirty nine genes were increased and 1078 were reduced in WT as compared to *Vhl cKO* CD4 T cells after anti CD3/CD28 stimulation (Supplementary Fig. 8e). Low numbers of differentially expressed genes were detected when comparing WT vs *Vhl Hif1a dcKO* CD4 T (Fig. 9h, i and Supplementary Fig. 8e). Among the 45 highest ranked differentially expressed transcripts in *Vhl cKO* vs WT CD4 T cells, 11 play a role in glycolysis, 8 are immune regulators, 5 are involved in oxygen sensing and 4 are inhibitors of proliferation (Supplementary Table 1). The KEEG pathway enrichment analysis showed that genes in the HIF-1 signaling and carbon metabolism pathways were enriched in TCR stimulated *Vhl cKO* as compared WT CD4 T cells, while genes involved in the positive regulation of cell cycle, mitosis and Rho GTPase activation were increased in WT as compared to *Vhl cKO* CD4 T cells (Fig. 9j, k). Upregulated genes in the HIF-1 pathway play a role in glycolysis, oxygen sensing, mitochondrial autophagy, cell cycle inhibition, angiogenesis and immune responses (Supplementary Fig. 8f). The GO analysis identified T cell activation, cytokine production, response to oxygen processes and dioxygenases enriched in anti-CD3/CD28 stimulated *Vhl cKO* as compared to WT CD4 T cells (Supplementary Fig. 8g, i). Genes involved in cell cycle/mitosis were increased in WT vs *Vhl cKO* CD4 T cells (Supplementary Fig. 8h, j).

The phosphorylation of the ribosomal protein 6 (rpS6) is a key component of the translational machinery in eukaryotic cells and a point of convergence for multiple signaling pathways downstream of the TCR[34]. While the frequency of WT, *Vhl cKO* and *Vhl Hif1a dcKO* CD4 T cells expressing phospho-rpS6 after anti-CD3/CD28 stimulation was similar the expression levels were reduced in *Vhl cKO* as compared to WT and *Vhl Hif1a dcKO* CD4 T cells (Fig. 9l, m and Supplementary Fig. 8k). Instead, the levels of phospho-rpS6 expression in WT and mutant CD4 T cells after PMA/I stimulation were similar (Fig. 9l), suggesting that PMA/I activation bypasses the HIF-1-mediated inhibition of a TCR proximal signaling. In line with this, the frequency of IFN-γ-secreting *Vhl cKO* CD4 T cells after anti-CD3/CD28 stimulation was lower than those from WT or *Vhl Hif1a dcKO* CD4 T cells, while the frequency of IFN-γ secreting *Vhl cKO*, *Vhl Hif1a dcKO* and WT CD4

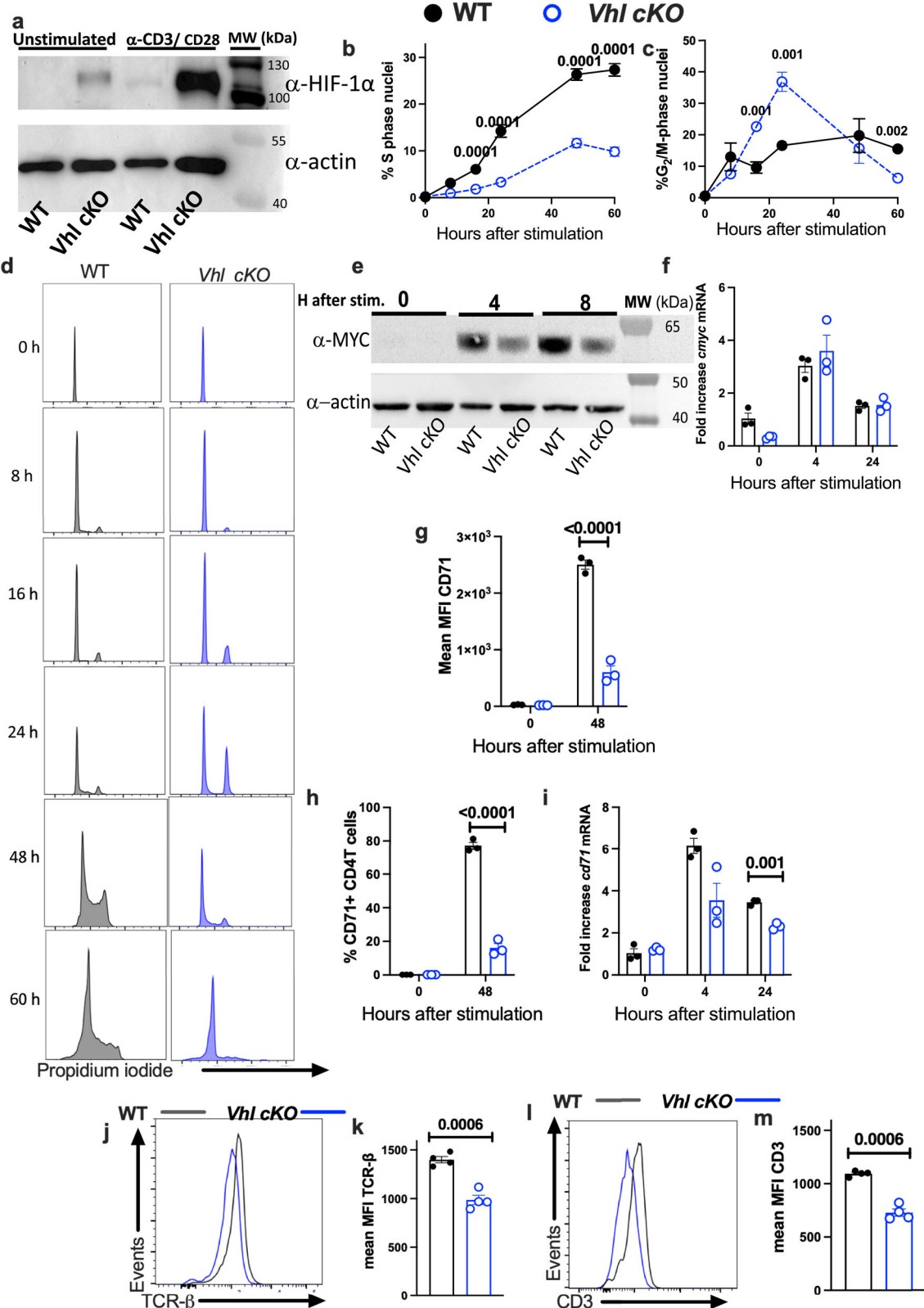

T cells stimulated with PMA/I was similar (Fig. 9n), also indicating that T cell activation in response to PMA/I is not impaired in *Vhl cKO* CD4 T cells. In agreement with the suggestion that TCR- activation is impaired upon Vhl-loss, the levels of CD44, CD69 and CD25 levels as well as the proliferative responses of *Vhl cKO* CD4 T cells were diminished in response to stimulation with the Staphylococcal enterotoxin B superantigen (SEB) (Supplementary fig. 9a–k). Instead, the fraction of CD44, CD69 and CD25 expressing WT and *Vhl cKO* T cells stimulated with PMA/I were similar (Supplementary fig. 9l–n).

## Discussion

Here, we report that stabilization of HIF-1 in T cells dramatically increased susceptibility to infection with *M. tuberculosis* that associated with a defective accumulation of mycobacteria-specific T cells in

**Fig. 5 | VHL expression is required for TCR activation of CD4 T cells. a** Naïve CD4 T cells were enriched by negative selection from *Vhl cKO* and WT spleens and stimulated with anti-CD3/CD28. Cell lysates were analysed by Western blot before and after TCR stimulation using anti-HIF-1α and anti-β-actin antibodies. One of three independent similar experiments is shown. **b–d** The mean % of S-phase (**b**) and G2/M (**c**) CD4 T cells and representative histograms (**d**) of the cellular DNA content after propidium iodide staining of *Vhl cKO* and WT CD4 T cells at different times after anti-CD3/CD28 stimulation are shown. **e** Cell lysates from *Vhl cKO* and WT CD4 T cells were analysed by immunoblot after anti-CD3/CD28 stimulation using anti-MYC or anti-β-actin antibodies. One representative of two independent experiments is shown. **f** Total RNA was extracted from *Vhl cKO* and WT CD4T cells at different times points after anti-CD3/CD28 stimulation. *Cmyc* mRNA titer was normalized to *Hprt* mRNA levels in the same samples. The *Cmyc* mRNA fold increase levels are depicted. **g, h** The frequency (**h**) and expression level (**g**) of CD71 in *Vhl cKO* and WT CD4T cells are shown. **i** The fold change levels of *cd71* mRNA in *Vhl cKO* and WT CD4 T cells was determined by real time-PCR. **j–m** Representative histograms and the MFI of the TCR-β (**j, k**) and CD3 (**l, m**) expression in *Vhl cKO* and WT CD4 T cells are shown. Each symbol represents one independent sample, and the data are presented as the mean ± SEM. n = 3 (**b, c, f–i**), and n = 4 (**k, m**). *p*-values were calculated using a two-way ANOVA (**b, c**) or a two-tailed unpaired *t* test with Welch's correction and a FDR adjustment (**f–i, k, m**). Source data are provided as a Source Data file.

the lungs, and the lack of ability of *Vhl cKO* CD4 T cells to confer resistance against infection upon transfer.

*M. tuberculosis* as many other pathogenic bacteria and viruses have evolved a strategy of persistence to optimize transmission[35]. A HIF-1 control of T cell functions and differentiation was not found to be an adaptation to allow mycobacterial evasion and chronic infection, since, after immunization with BCG, *Vhl cKO* mice also showed reduced accumulation of mycobacterial-specific CD4 and CD8 T cells, diminished CD4 T cell frequencies and low levels of the differentiation markers KLRG1 and CX3CR1 in lung and spleen CD4 and CD8 T cells and high levels of the inhibitor receptor CTLA-4.

Our in vivo and in vitro data disclosed that after TCR-stimulation, VHL-deficient CD4 T cells showed decreased growth response, altered cell cycle progression, reduced proliferation and increased expression of apoptotic markers, responses that were regulated by the transcription factor MYC. MYC, like HIF-1, has pleiotropic effects on metabolism, proliferation, and cell growth of different cell populations including T cells. In T cells, MYC is regulated by TCR and IL-2 signaling and supports T cell differentiation[36]. MYC also regulates IL-2 responses but adding IL-2 to *Vhl cKO* CD4 T cells did not restore the proliferative responses after TCR stimulation, suggesting that the defect in IL-2 generation alone is not solely responsible for the activation defects of *Vhl cKO* CD4 T cells. Studies of hypoxia-induced cell cycle arrest have demonstrated that HIF-1 counteracts the effects of MYC on proliferation, although HIF-1 and MYC can also have synergistic effects on angiogenesis and metabolic responses[15,37]. The overexpression of HIF-1 has been shown to displace MYC from its DNA binding sites, leading de-repression of the genes coding for cyclin dependent kinases (CDK) inhibitors p21 and p27[37,38]. HIF-1 has been also shown to mediate regulation of MYC via the induction of the MYC-antagonist MXI-1[15,39] and the promotion of the proteasome-dependent degradation of MYC[15]. Others have shown that HIF-1α can directly interact with Mcm replication proteins to regulate cell cycle progression[40]. Accordingly, *mxi1*, *cdnk1a* and *cdkn1b* transcripts were increased in *Vhl cKO* CD4 T cells in the lungs of *M. tuberculosis*-infected mice, where genes involved in proliferation, cell cycle progression and mitochondrial activity were downregulated.

We suggest that Vhl-deficiency required for early molecular events after TCR-mediated activation after anti-CD3/CD28 or SEB-stimulation. Bypassing direct TCR signaling with PMA/ionomycin normalized the early *Vhl cKO* CD4 T cell responses. Homeostatic proliferation of CD4 and CD8 T cells has been shown to require the contact with self-MHC class II and I molecules, respectively[41–43]. Thus, the deficient *Vhl cKO* CD4 T cells reconstitution in blood and spleen from *Rag2−/−* lymphopenic mice also supports a TCR-signaling defect of the mutant cells.

The loss of HIF-1 in CD4 T cells has been shown to compromise help during antibody responses, and the ability of both CD4 and CD8 T cells to produce IFN-γ[44] HIF-1 has also been shown to enhance Th17 as well as IFN-γ secretion and attenuate the development of Treg cells[45–48]. The deficiency of VHL in CD8 T cells resulted in improved control of persistent viral infection and neoplastic growth in a HIF-

dependent manner[16,18]. VHL deficiency resulted in a sustained CD8 T cell effector phenotype and T cell activation[16,18], without impairing the formation of memory cells[49]. In line with this, the loss of PHD proteins expanded $T_H1$ responses, limited Treg induction, and enhanced CD8 T cell effector functions[17].

In contrast to these observations, hypoxia has been shown to inhibit the proliferation of numerous cell types in a HIF-1-dependent manner[38,50], and overexpression of HIF-1 or HIF-2 alone was sufficient to induce cell cycle arrest in normoxic conditions[51]. T cells activated under hypoxic conditions showed reduced proliferation, impaired effector functions and increased cell death[52–55]. The inhibition of glycolysis (and HIF-1 expression) enhanced memory-like CD8 T cells and artificially enforcing glycolytic metabolism restricted the generation of memory CD8 T cells[56]. In line with our results, VHL expression in mature T cells has been shown to promote follicular helper T cell and $T_H17$ differentiation[48,57].

The expression of inhibitory receptors CTLA-4, PD-1, *Cd73, Nr4a2* and *Nr4a3* in *Vhl cKO* CD4 T cells was increased while transcription factors and molecules involved in T cell differentiation (T-bet, eomes, CX3CR1 and KLRG1) were downregulated in *Vhl cKO* CD4 T cells, during *M. tuberculosis* infection. A reduction of CD4, but not of CD8 T cells frequencies, a high expression of inhibitory molecules and low levels of chemokine receptors were observed in the lungs and spleens of *M. tuberculosis*-infected or immunized *Vhl cKO* mice. Lungs and spleens from *Vhl cKO* infected, immunized and control mice, showed a higher frequency of CD44 + CD62L + CD8 T cells than WT mice. In non-immunized mice these cells lacked CD49d expression suggesting that they are $T_{VM}$ cells, antigen-naïve T cells that bear markers of homeostatic expansion[30,58]. After infection, the lung *Vhl cKO* CD8 $T_{CM}$ showed low levels of TB10.4 tetramer positive cells as well as low CD49d expression. In normal mice $T_{VM}$ have been shown to comprise 5-20% CD8 T cells, and have been shown to preferentially differentiate into CD8 $T_{CM}$[30,58]. Virtual memory CD8 T cells have been shown to display low IFN-γ responses but could productively contribute to antigen-specific responses against invading viruses[59,60] and bacteria[61], and can account for the discrepancies indicated above. The fraction of *Vhl cKO* CD44 + CD62L + CD4 T cells in lungs or spleens was in comparison low.

VHL mediates the proteasomal degradation of both HIF-1α and HIF-2α subunits. HIF-2 has both distinct and overlapping biological roles with HIF-1. HIF-1 has been shown to induce apoptotic pathways and drive the expression of genes that are involved in the glycolytic pathway, whereas HIF-2 preferentially promotes growth and angiogenesis[30,62]. Whereas HIF-1 is ubiquitously expressed, the expression of HIF2 is mainly restricted to endothelial, lung, renal and hepatic cells[63]. The ablation of the *Vhl* gene increased the expression of HIF-2 in CD8 T cells[16] and HIF-2 deletion in T regs has been shown to impair their suppressive function (through the increase of HIF-1 levels)[64]. Ectopic expression of HIF-2α, but not HIF-1α increased cytotoxic, differentiation and cytolytic function against tumor targets of CD8 T cells[65]. VHL has also HIF-independent functions, regulating apoptosis, cell senescence and transcription among other programs[66,67]. Our data indicates that impaired T cell-mediated

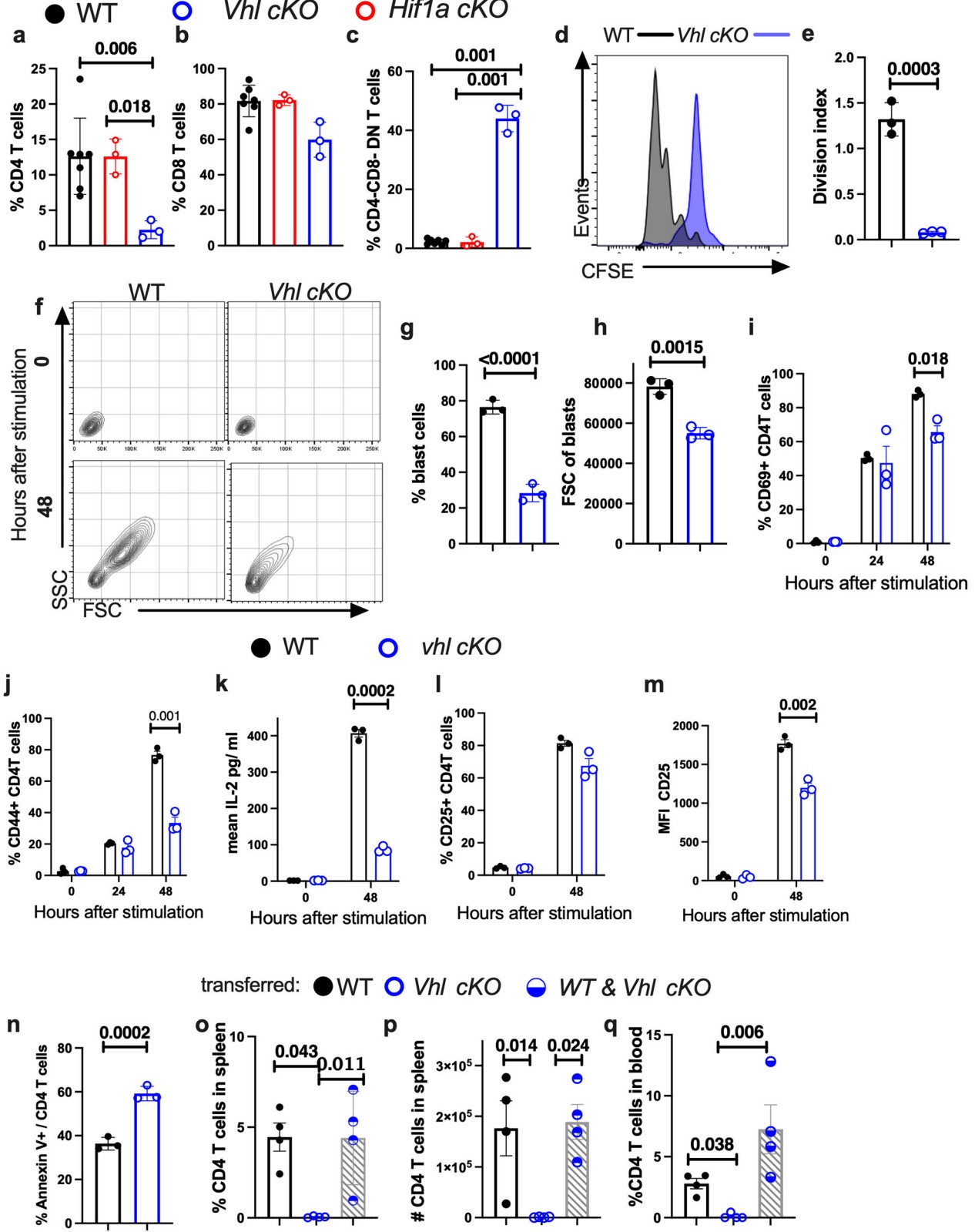

protection against *M. tuberculosis* in *Vhl cKO* mice was due to stabilization of HIF-1. Redundant gene regulation by HIF-2 or the HIF-independent functions of VHL did not account for the dysfunctional CD4 T cells in *Vhl cKO* observed. Moreover, the altered metabolic responses in *Vhl cKO* CD4 T cells were restored when in *Vhl Hif1a dcKO* mice.

Thus, we showed HIF-1 play impairs CD4 T cell activation and differentiation leading to an increase susceptibility to infection with *M. tuberculosis* and impaired responses to vaccination (Fig. 9o). While this may have ample consequences in diverse microbial insults, we suggest that the hypoxic environment of the TB granuloma, by impairing local CD4 T cell functions[68], may contribute to bacterial survival and chronicity of infection.

**Fig. 6 | VHL promotes proliferation of CD4 T cells in response to TCR activation. a–c** The % CD4 (**a**), CD8 (**b**) and DN (**c**) T cells in *Vhl cKO* ($n = 3$), *Hif1a cKO* ($n = 3$) and WT ($n = 6$) spleen cell suspensions measured 6 days after anti-CD3/CD28 stimulation is shown. **d** Representative histogram of CFSE-labeled *Vhl cKO* and WT CD4 T cells 3 days after stimulation with anti-CD3/CD28 are shown. **e** The CFSE labeling was used to calculate the division index (the average number of cell division that a cell in the original population has undergone). **f–h** Dot plots showing the FCS/SSC of lymphocyte gated *Vhl cKO* and WT CD4 T cells before and 48 h after stimulation with anti-CD3/CD28 (**f**). The mean percentage of blast cells (**g**) and the FSC of blasts (**h**) 48 h after anti-CD3/CD8 stimulation are shown. **i, j** The percentage of CD69+ (**i**) and CD44 + (**j**) *Vhl cKO* and WT CD4 T cells after anti-CD3/CD28 stimulation are depicted. **k** The titers of IL-2 in the culture supernatants from *Vhl cKO* and WT CD4 T cells before and 48 h after anti-CD3/CD28 stimulation. **l, m** The percentage of cells expressing CD25 (**l**) and the MFI of CD25 (**m**) on TCR-stimulated *Vhl cKO* and WT CD4 T cells is depicted. **n** The percentage of Annexin V + *Vhl cKO* and WT CD44 + CD4 T cells 3 days after stimulation with anti-CD3/CD28. **e**, **g–n** $n = 3$ independent cultures per group. **o, p** *Rag2$^{-/-}$* mice were transferred i.v. with either $2.10^6$ CD4 T cells from CD45.2 *Vhl cKO*, CD45.1 WT or a mixed 1:1 suspension from both genotypes ($10^6$ cells/each). The percentages and numbers of CD4 T cells in the spleen (**o**, **p**) and blood (**q**) of mice 5 weeks after transfer is shown ($n = 4$ mice per group). Each symbol represents one independent biological sample, and the data are presented as the mean ± SEM. *p*-values were analysed using two-tail unpaired *t* test (**e**, **g–n**) with FDR adjustment or one-way ANOVA with Welch's correction (**a–c**, **o–q**). Source data are provided as a Source Data file.

## Methods

### Mice

The animals were housed according to directives and guidelines of the Swedish Board of Agriculture, the Swedish Animal Protection Agency, and the Karolinska Institutet (djurskyddslagen 1988:534; djurskydds-förordningen 1988:539; djurskyddsmyndigheten DFS 2004:4). The study was performed under approval of the Stockholm North Ethical Committee on Animal Experiments permit number 1374-2020 and N128/16.

Mice were housed at the Comparative Medicine Biomedicum and the Astrid Fagræus Laboratories, Karolinska Institutet, Stockholm, under specific pathogen-free conditions. All mice in this study were between 8–15 week/old. Mice were maintained in a specific pathogen free unit on a 12 h light/12 h dark cycle. Room temperature was maintained at 25 °C. The humidity level was controlled between 40–60%.

Mice containing loxP-flanked *Hif1a* and *Vhl* alleles have been previously described[69,70]. For a T-cell-specific deletion, these were bred with transgenic mice containing *Cre*-recombinase gene driven by the distal promoter of the lymphocyte protein tyrosine kinase (*dlck cre*)[71] or under the CD4 promoter (*cd4 cre*)[72] both resulting in cre expression during the DP stage thymocyte development. *Hif1a$^{fl/fl}$ or Vhl$^{fl/fl}$* littermates were used as controls. *Vhl$^{fl/fl}$* and *hif1$^{fl/fl}$ cd4 cre* mice were also crossed to generate mice lacking both *Hif1a* and *Vhl* genes in T cells. The C57BL/6 (Ly5.1 B6) congenic strain carrying the differential pan leukocyte marker CD45.1 and T and B-cell deficient *Rag2$^{-/-}$* mice with a C57Bl/6 background were used in T-cell co-transfer mice studies.

### Infection and infectivity assay

*M. tuberculosis* Harlingen and BCG Montreal were grown in Middlebrook 7H9 (Difco, Detroit, MI) supplemented with albumin, dextrose, catalase and, for BCG cultures, 50 µg/ml hygromycin (Sigma, St. Louis, MO). Mice were infected with 250 *M. tuberculosis* Harlingen strain by aerosol using a nose-only exposure unit (In-tox Products, Moriarty, NM), or immunized i.v. with $10^7$ BCG.

To determine viable numbers of *M. tuberculosis* and BCG CFUs at different time-points post-infection, the right lung of each mouse was homogenized in PBS with 0.05% Tween 80. Bacteria were quantified on Middlebrook 7H11 agar containing 10% enrichment of oleic acid, albumin, dextrose, catalase, 5 µg of amphotericin B per ml and 8 µg/ml polymyxin B grown for 3 weeks at 37 °C.

### Naïve CD4 T cell isolation and activation in vitro

Single cell suspensions were obtained from spleens of mice by mechanical disruption, filtering over a 70 µm nylon cell strainer and lysis of erythrocytes. Label-free naïve CD4$^+$CD44$^-$ T cells were further isolated by negative selection using magnetic beads (MACS, Miltenyi Biotec, Germany). After counting, splenocytes were resuspended in PBS with 0.5% bovine serum albumin (BSA) and 2 mM EDTA, incubated with biotin-conjugated anti-CD44 microbeads and applied through a separation column under a magnetic field. Selected cells were then resuspended in RPMI-1640 media supplemented with 5% FCS and penicillin/streptomycin stimulated in vitro with either plate-bound anti-CD3 antibody (145-2C11, Invitrogen, Waltham, MS) and soluble anti-CD28 antibody (37.51, BD, Franklin Lakes, NJ), with 50 ng/ml phorbol myristate acetate (PMA) and 2 µg/ml ionomycin (Sigma, StLouis, MO), or with 20 µg/ml Staphylococcal enterotoxin B (SEB) (Sigma) and cultured at $CO_2$ incubator at 37 °C and 5% $CO_2$.

### Flow cytometry

**Surface markers.** Lungs were removed, mechanically minced into small pieces and digested with 3 mg/ml Collagenase D and 30 µg/ml DNase I for 1 h at 37 °C, and single-cell suspensions prepared by filtering lung tissue through 70 µm nylon cell strainers. To enrich the suspension in lymphocytes, cells were loaded into an isotonic 40–70% Percoll density gradient and centrifuged for 30 min at room temperature. Cells in the gradient interphase were collected and washed before further culturing/labeling. Mediastinal lymph node cell suspensions were obtained after mechanical disruption of the followed by filtering over a 70-µm nylon mesh. Single spleen cell suspensions were obtained by mechanical disruption, lysis of erythrocytes and straining over a 70-µm nylon mesh. Cell suspensions were incubated with live/dead stain (LIVE/DEAD™ Fixable Yellow Dead Cell Stain, Invitrogen). Then CD16/CD32 blocking antibodies (BD) and the fluorophore conjugated antibody cocktails (Supplementary Table 2) were introduced and incubated for 30 min on ice. Cells were then washed with PBS, resuspended and fixed with 2% paraformaldehyde solution in PBS. Data were acquired on a LSRII or a FACS Canto II flow cytometers controlled by a FACSDiva software version 6.0 and analyzed with FlowJo v10.7 (Tree star Inc., Ashland, OR). Examples of the gating strategies used are shown (Supplementary Fig. 10).

**Intravascular staining.** In some experiments, tissue-localized and blood-borne cells were discriminated by intravascular staining[73]. In short, mice were inoculated i.v. with 3 µg of FITC-labeled anti-CD45.2 (clone 104 BD), sacrificed 3–5 min after i.v. inoculation and lung cell suspensions were obtained and obtained as described above. Peripheral blood was sampled for every mouse as a positive control for i.v. labeling.

**Tetramer staining.** MHCII tetramers containing amino acids 1–20 of *M. tuberculosis* ESAT-6 or 240–254 of Ag85B and the MHCI tetramer containing amino acids 4–11 TB10.4 (all from the NIH Tetramer Core Facility, Atlanta, GA) were used for detection of *M. tuberculosis*-specific murine CD4 or CD8 T cells. Single-cell lung or MLN suspensions were stained at saturating concentrations with the tetramers and incubated at 37 °C for 1 h for the MHCII tetramers and at 4 °C for 30 min for the MHCI tetramer. Cells were then washed and stained with the fluorochrome antibody cocktails as described above.

**Intracellular staining.** For determination of IFN-γ-producing cells, spleen CD4 T cells were stimulated with either PMA/Ionomycin or anti-CD3/CD28, and lung cell suspensions from *M. tuberculosis*-infected mice were incubated with either 5 µg/ml ESAT6$_{1-15}$, 5 µg/ml TB10.4$_{4-11}$ or PMA/Ionomycin for 6 h at 37 °C. Brefeldin (10 µg/ml) was added to

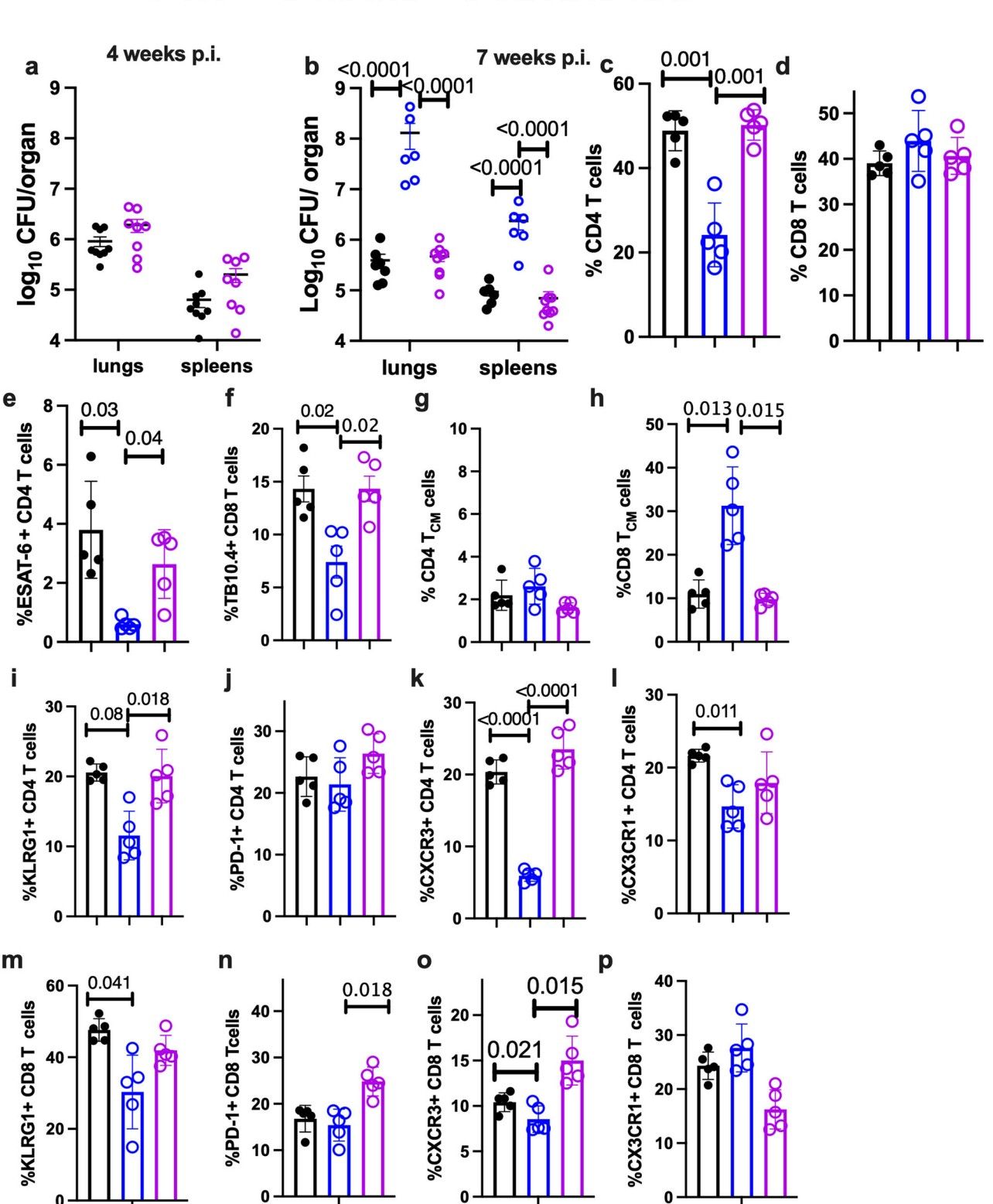

**Fig. 7 | HIF-1 mediates the susceptibility to *M. tuberculosis*-infection of *Vhl cKO* mice. a, b** *Vhl cKO*, *Vhl Hif1a dcKO* and WT mice were sacrificed at 4 (**a**) and 7 (**b**) weeks after aerosol infection with *M. tuberculosis*. The log$_{10}$ CFU in lungs and spleens are depicted; (**a**) WT $n = 9$, *Vhl Hif1a dcKO* $n = 8$, (**b**) WT $n = 6$, *Vhl cKO* $n = 6$, *Vhl Hif1a dcKO* $n = 9$. **c, d** The frequency of CD4 (**c**) and CD8 (**d**) T cells in the lung of *Vhl Hif1a dcKO*, *Vhl cKO* and WT mice 7 weeks after *M. tuberculosis* infection is shown. **e, f** The percentage of tetramer ESAT-6-binding CD4 T cells (**e**) and TB10.4-binding CD8 T cells (**f**) in the lung of *Vhl Hif1a dcKO*, *Vhl cKO* and WT mice 7 weeks after infection are displayed. **g, h** The frequency of CD4 (**g**) and CD8 (**h**) T$_{CM}$ in the lung of *Vhl cKO*, *Vhl Hif1a dcKO* and WT mice 7 weeks after *M. tuberculosis* infection is shown. **i–p** The frequency of KLRG1 + (**i, m**), PD-1 + (**j, n**), CXCR3 + (**k, o**) and CX3CR1 + (**l, p**) CD4 and CD8 T cells in the lung of *Vhl Hif1a dcKO*, *Vhl cKO* and WT mice 7 weeks after infection with *M. tuberculosis*. **c–p** $n = 5$ mice per group. Each symbol represents one mouse, and the data are presented as the mean ± SEM. *p*-values were calculated using one-way ANOVA test with Welch's correction for unequal SD and Dunnet's T3 adjustment for multiple comparisons. Source data are provided as a Source Data file.

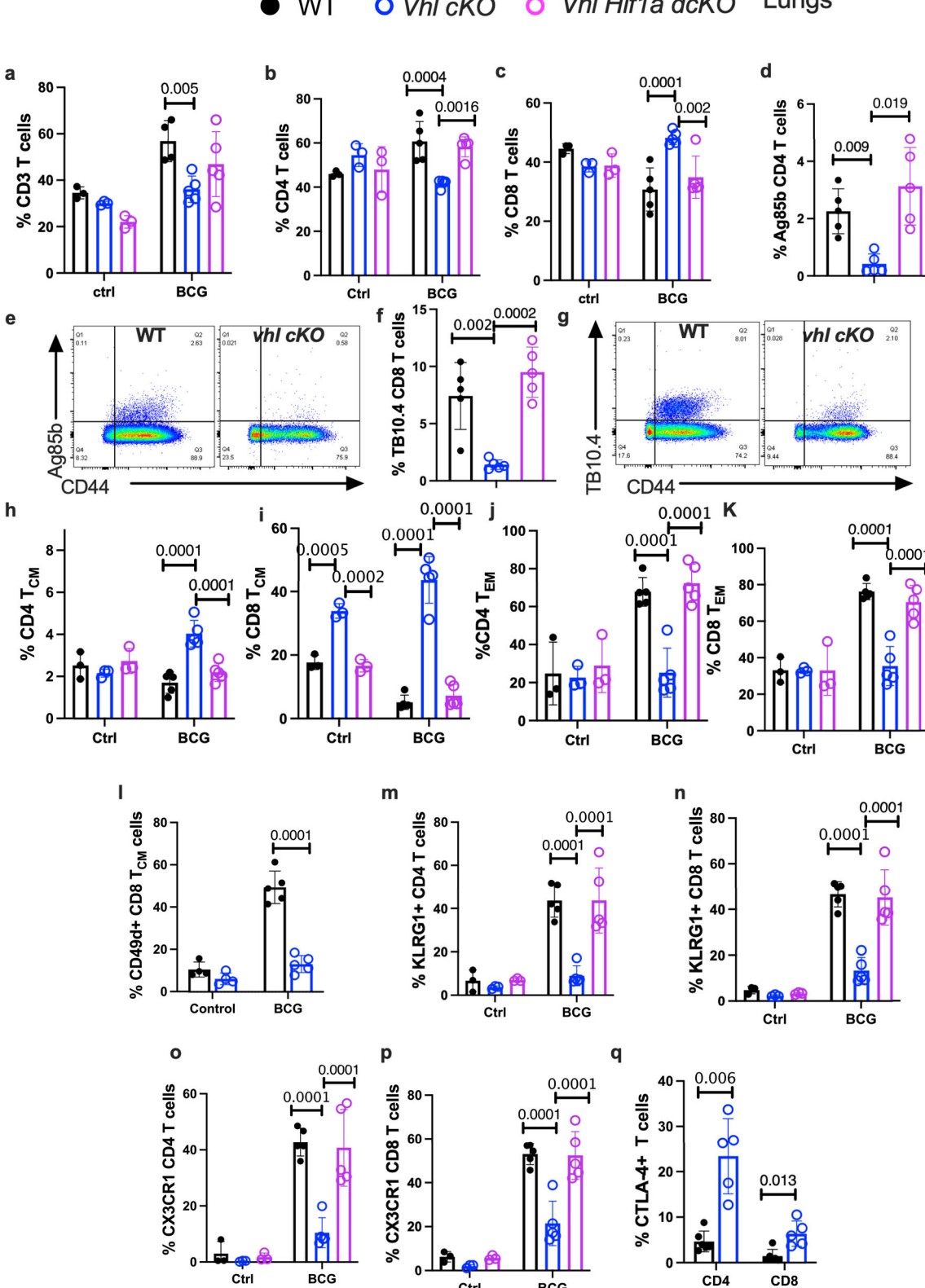

**Fig. 8 | HIF-1 stabilization in T cells impairs the responses to immunization with BCG. a–c** The frequency of total (**a**), CD4 (**b**) and CD8 (**c**) T cell populations in lungs before and 3 weeks after i.v. immunization with BCG are depicted. **d–g** The percentages (**d, f**) and representative dot plots (**e, g**) of tetramer Ag85b-binding CD4 T cells (**d, e**) and TB10.4 tetramer-binding CD8 T cells (**f, g**) in the lungs of *Vhl Hif1a dcKO, Vhl cKO* and WT mice (*n* = 5 per group) after BCG immunization are shown. The frequencies of T_CM (**h, i**), T_EM (**j, k**) in CD4 and CD8 T cells in lungs from *Vhl Hif1a dcKO, Vhl cKO* and WT mice at 0 or 3 weeks after BCG immunization are depicted. **l** The percentage of lung CD49d + CD8 T_CM before and after BCG immunization of WT and *Vhl cKO* mice are shown. The frequencies of KLRG1+ (**m, n**) and CX3CR1+ (**o, p**) in CD4 and CD8 T cells in lungs from *Vhl Hif1a dcKO, Vhl cKO* and WT mice at 0 or 3 weeks after BCG immunization are depicted. **q** The fraction of CTLA-4 + CD4 and CD8 T cells in the lung of BCG immunized mice are shown. Each symbol represents one mouse, and the data are presented as the mean ± SEM (**a–k, m–p**) before immunization *n* = 3 per mice per group, after immunization *n* = 5 per group; **l** before *n* = 4 per group, after immunization *n* = 5 per group; **q** *n* = 5 per group. *p*-values were calculated using 2-way ANOVA test, with Sidak adjustment for multiple comparisons. Source data are provided as a Source Data file.

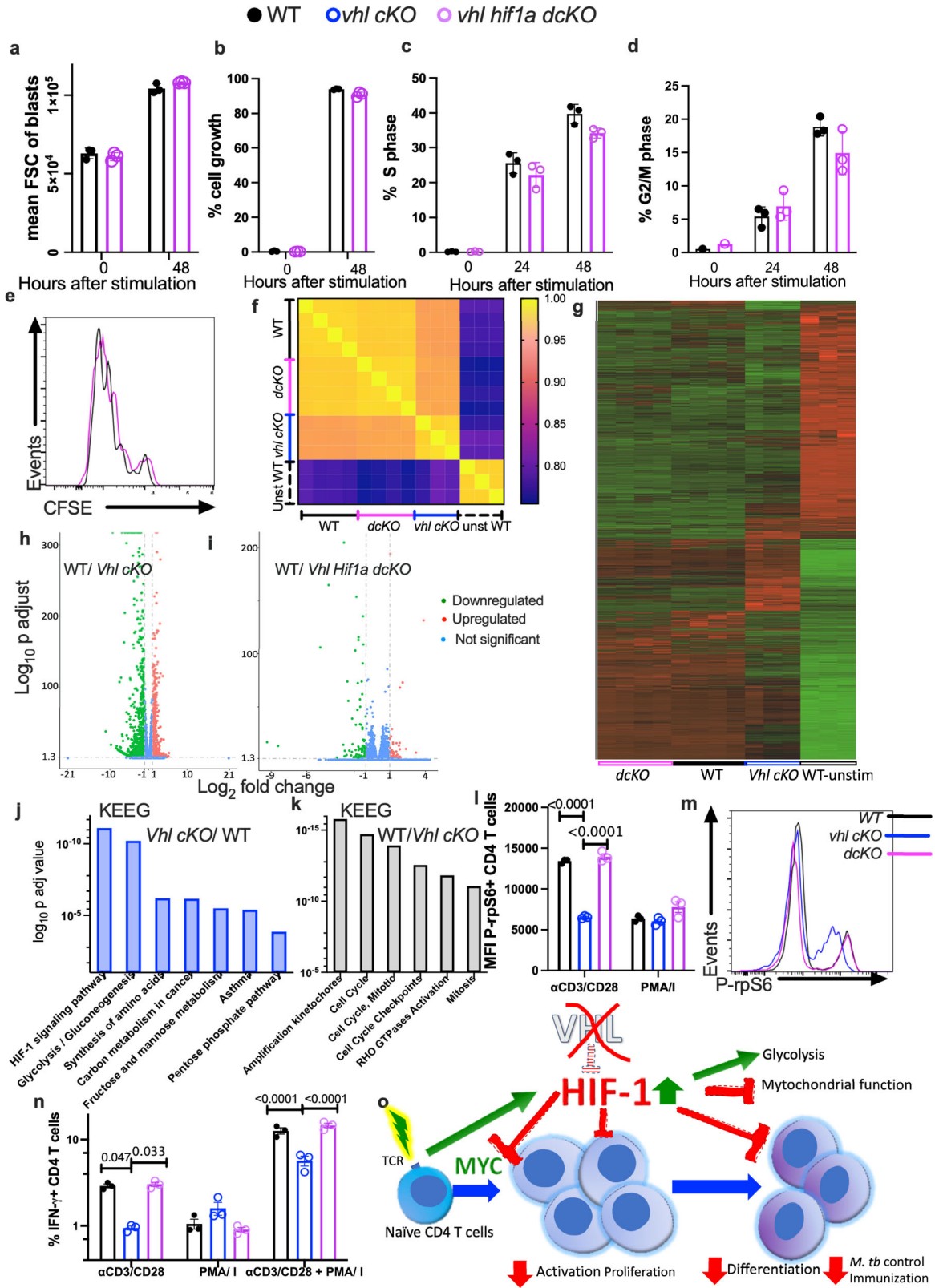

the cultures the last 4 h of stimulation. The IFN-γ secretion in naïve cells were then stained with live/dead staining followed by cell population-specific antibody cocktails. After labeling, cells were subsequently fixed and permeabilized using the leukocyte permeabilization reagent kit IntraPrep™ (Beckman, Brea, CA) and further stained with anti-IFN-γ (eBioscience, San Diego, CA). Data were acquired as described above.

To determine the expression of CTLA-4, Foxp3 and Ki-67, cell suspensions were prepared as described above were fixed and permeabilized using the eBioscience FOXP3/Transcription Factor Staining Buffer Set (Invitrogen) according to the manufacturer's protocol and stained with specific antibodies for FOXP3, Ki-67 and CTLA-4.

To evaluate the expression of phospho-ribosomal protein S6, CD4 T cell suspensions were stained for surface markers as above, fixed

**Fig. 9 | HIF-1 stabilization impairs CD4 T cell responses in vitro. a, b** The FSC of blasts (**a**) and the % cell size increase relative to unstimulated cells in the blast population (**b**) were depicted. **c, d** The % of S-phase (**c**) and G2/M (**d**) of CD4 T cells are shown. **e** CFSE profiles of *Vhl Hif1a dcKO* and WT CD4 T cells 3 days after stimulation with anti-CD3/CD28 are shown. **f**–**m** RNA-seq was performed in independent cultures of WT (*n* = 4), *Vhl cKO* (*n* = 3) and *Vhl Hif1a dcKO* CD4 T (*n* = 4) cells before or 24 h after anti CD3/CD28 stimulation. **f** The Pearsson $R^2$ correlation matrix of the expression of all genes in each sample is shown. **g** Heat map of the $\log_2$ counts for each gene standardized to the mean. **h, i** Volcano plots showing the $\log_2$ fold change in gene expression (*x*-axis) and *p*-value (*y*-axis), of differentially upregulated or downregulated genes (*p* ≤ 0.05 and $\log_2$ fold change > 1) (**i**). **j, k** The $\log_{10}$ *p* value of the most enriched KEEG-pathway terms in *Vhl cKO* vs WT (**j**) or in WT vs *Vhl cKO* CD4 T cells (**k**). **l, m** The MFI (**l**) and a representative histogram (**m**) of phospho-rpS6 on *WT* and mutant CD4 T cells 2 h after anti-CD3/CD28 or PMA/I stimulation are illustrated. **n** The frequency of IFN-γ expressing WT, *Vhl cKO* and *Vhl Hif1a dcKO* CD4 T cells 6 h after PMA/I or 72 h after anti-CD3/CD28 stimulation are shown. A group was stimulated with anti-CD3/CD28 for 65 h and then incubated for 6 h with PMA/I. **a**–**d, l, n** Each symbol represents one independent sample (*n* = 3 per group), and the data are presented as the mean ± SEM. *p*-values were calculated using a two-tailed unpaired *t* test with Welch's correction (**a**–**d**) and 2-way ANOVA test, with Sidak adjustment for multiple comparisons (**l, n**). Source data are provided as a Source Data file. **o** VHL promotes the cell cycle progression, growth responses, proliferation, IL-2 secretion and expression of activation markers of TCR-stimulated CD4 T cells, the differentiation and the protective function of T cells against *M. tuberculosis* infection as well as the responses to immunization by impairing HIF-1 stabilization.

with 2% PFA for 30 min on ice followed by permeabilization with Perm Buffer III (BD) for 30 min on ice.

**CFSE proliferation assay.** Cell suspensions were labeled in vitro with carboxyfluorescein succinimidyl ester (CFSE), to monitor distinct generations of proliferating cells by dye dilution. Prior to seeding, cells were labeled with 5 μM CFSE in PBS for 7 min and the reaction was stopped with 2 ml of ice cold 1% fetal calf serum (FCS) to absorb any unbound dye.

**Cell cycle profile and apoptosis assay.** To determine the cell cycle profile DNA was stained with propidium iodide. CD4 T cells were harvested, briefly vortexed to create a single cell suspension and fixed with 70% ethanol for 2 h on ice. Fixed cells were resuspended in PBS, washed twice, and treated with 100 μg/ml RNase A (Sigma) and 50 μg/ml propidium iodide before acquisition on the flow cytometer.

Apoptosis was determined by Annexin V binding using FITC Annexin-V Apoptosis Detection Kit I (BD) according to supplier's protocol. Cell suspensions were stained with surface markers and then incubated with 5 μL of PI and the Annexin V kit for 15 min at room temperature before analysis on the flow cytometer.

**Western blot**
CD4 T cells were sorted from lung cell suspensions from *M. tuberculosis*-infected mice by positive selection using MACS CD4 (L3T4) microbeads (Miltenyi Biotech). CD4 T cells either from infected mice or after stimulation in vitro with anti-CD3/CD28 were lysed in in RIPA buffer supplemented with protease and phosphatase inhibitor cocktail. Protein concentration was measured using BCA protein assay kit (Biorad) and 15 μg protein were and separated on 10% separating/5% stacking SDS-polyacrylamide gels. Samples were then transferred onto nitrocellulose membranes (BioRad, Hercules, CA) by electroblotting at 100 V, 250 mA for 80 min. Immunostaining was performed using polyclonal rabbit anti-MYC or anti-HIF-1α or anti-actin (see Supplementary Table 3 for antibodies used). Membranes were then washed and incubated with horse-radish peroxidase-conjugated polyclonal goat anti-rabbit immunoglobulin and developed using ECL-Plus (Amersham Biosciences, Buckinghamshire, UK) and photographed using a Fuji intelligent dark box II digital camera.

**Histopathology**
The left lungs of mice infected with *M. tuberculosis* were dissected, fixed in 4% paraformaldehyde and embedded in paraffin. From each lung sample 4 15 μm sections were obtained and stained with hematoxylinn-eosin before evaluation.

**Real time-PCR**
Selected transcripts were quantified in lysates from CD4 T cells before or after in vitro anti CD3/CD28-stimulation by real time PCR as previously described[74]. *Hprt* was used as a control gene to calculate the $\Delta C_t$ values for independent triplicate samples. The primer sequences used are displayed in Table S2. The relative amounts of target/*hprt* transcripts were calculated using the $2^{-(\Delta\Delta Ct)}$ method. These values were then used to calculate the fold increase of specific mRNA in CD4 T cells.

**RNA sequencing**
RNA was extracted from CD4 T cells obtained by positive selection from lungs of *M. tuberculosis* infected mice using CD4 Macs beads (Milteny) or from cultures of splenic CD4 T cells 24 h after stimulation with anti-CD3/CD28. RNA was isolated using miRNeasy micro kit (QIAGEN, Hilden, Germany) according to the manufacturer's instructions and processed for sequencing at bioinformatics and expression analysis core facility (BEA) at Karolinska Institutet and at Novogene Biotech Co (UK). The RNA quality was assessed by 2200 TapeStation Instrument (Agilent, Santa Clara, CA). PolyA RNA selection was performed using the Illumina TruSeq RNA Sample Preparation Kit according to the manufacturer's protocol. RNA-seq libraries were prepared and sequenced on the Illumina HiSeq 2000 platform at Bioinformatics and Expression Analysis core facility (BEA, Karolinska Institutet, Sweden).

Preprocessed reads were aligned to the standard mouse reference genome mm10 using the HISAT2 program, and Hypergeometric Optimization of Motif EnRichment (HOMER, http://homer.salk.edu/homer) was used to create the tag directory and count tags in all exons. For the gene expression analysis, unsupervised hierarchical clustering and principal component analysis of genes were performed in Qlucore Omics Explorer 3.2 (Qlucore, Lund, Sweden). Differentially expressed genes were determined by comparing groups using heteroscedastic two-tailed t tests. Multiple testing correction was performed using the Benjamini-Hochberg algorithm with a false discovery rate (FDR) of 1%. Gene Ontology enrichment analysis (Biological Process, Molecular Functions) or KEEG pathway enrichment was performed with Web-Gestalt (http://www.webgestalt.org) using default parameters. The raw and processed RNA sequencing data can be accessed at the GEO accession numbers GSE190791 and GSE190909.

**Statistics**
Statistical analysis and graphical representation of data were done using GraphPad Prism 9 software version 3(GraphPad Prism, San Diego, CA). We have used the unpaired *t*-test with Welch's correction which assumes normal distribution but can be used when the two samples have unequal variances, and used a False discovery rate approach for multiple comparisons. We have two way ANOVA with a Sidak correction for multiple comparisons of the same parameters (for example for analyzing kinetics). Bacterial titers were $\log_{10}$-transformed for normalization. Statistical significance between three or more groups was determined using one- or two-way ANOVA. The in vitro experiments are performed with triplicate biological replicates and each experiment was repeated at least twice. The bar plots showing the mean and error bars have symbols that denote the individual values.

## Reporting summary

Further information on research design is available in the Nature Research Reporting Summary linked to this article.

## Data availability

The raw and processed RNA sequencing data have been deposited in the publicly accessible Gene Expression Ommnibus (GEO) NIH database repository and can be accessed using GEO accession numbers GSE190791 and GSE190909. The authors declare that data supporting the findings of this study areavailable within the paper and its supplementary information files. Source data are provided with this paper.

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

## Acknowledgements

We thank the expert help of the staff of the Astrid Fagreus animal house, Dept of Comparative Medicine, Karolinska Institutet, and to Dr Juan Basile at the FACS facility Karolinska Institutet. We would like to thank the National Institutes of Health Tetramer Core Facility for providing reagents. We acknowledge the comments and suggestions from Dr Randall Johnsson (University of Cambridge and Karolinska Institutet), Dr Carmen Gerlach (Karolinska Institutet), and the help in investigations to Dr Maria Arslenian Henriksson, Dr Lourdes Sainero Alcolado, Eric Calvo Bizarro, Mahadevan Venkita Subramani and Asma Araba (Karolinska Institutet). This study was supported by the Swedish Heart and Lung foundation 2018-20/20170491, the Swedish Research Council 2019-01691 and 2019-04725, the Swedish Institute for Internationalization of Research (STINT) 4-1796/2014, the European Community H2020 EMITB (grant number 643558), the Chinese Scholarship Council and the Karolinska Institutet.

## Author contributions

R.L., V.M., W.M., H.L. designed and performed experiments and analyzed data, N.U., J.W., J.Y., M.A. performed experiments and analyzed the data, B.C., S.S., J.H. provided resources, analyzed data, B.C., L.G.L., and M.R. interpreted the data and wrote the manuscript.

## Funding

## Competing interests

The authors declare no competing interests.
