## [Peer Review File · Nature Communications]

HIF-1 stabilization in T cells hampers the control of
Mycobacterium tuberculosis infectionREVIEWER COMMENTS

Reviewer #1 (Remarks to the Author):

This study uses a mouse model of TB infection with a conditional T cell KO in VHL or Hif-1 (or both) to examine the effect of HIFs the immune response to Mtb infection. There are a series of systematic experiments that reveal a negative effect of VHL conditional KO in T cells in the control of Mtb infection, which appears to be mediated by the stabilization of Hif-1a. This study follows up several other studies, particularly on the role of HIF-1a, which are not conclusive and somewhat contradictory, thus providing additional information related to the complex role of HIFs in TB. The manuscript is well written and the story is easy to follow; the experimental design and data analysis are robust.

The first few panels of fig 1 are very clear and show the increased susceptibility of VHL conditional KO mice to Mtb infection, with elevated bacterial burdens decreased survival and weight and increased lung pathology. On the other hand, Hif-1a cKO display the same bacterial burdens as control mice.

From this point on, including the data from the next few figures, the message is not very clear, as it gets lost in so many panels showing mostly frequencies of different populations of T cells. Only in very few occasions, the number of cells is depicted. Unarguably, the frequency of total CD4 and CD8 T cells is relevant here (certainly before Mtb infection, to show that VHL or HIF-1a cKO mice have normal frequencies of T cells), but the numbers of cells are equally relevant to understand what do these reduced frequencies mean. For instance, the authors show a dramatic reduction of CD4 T cells in the lungs of VHL cKO mice after Mtb infection but is the number of these cells reduced as dramatically? If so, is this reduction in activated T cells specifically, or are both naïve and activated T cells affected equally? Based on these data and the next figure one would expect the Ag-specific population to be affected the most (if not uniquely). However, the authors show a 50% reduction in CD4 T cells in the MLN, and it is very unlikely that 50% of CD4 T cells in the MLN are Mtb-specific. This is important to clarify as the main message of the paper that suggest that the effects of VHL cKO is dependent on TCR activation (which is not clear, see below) but if this is indeed the case how can one explain the dramatic reduction of CD4 T cells in the MLN, at 8 wks post-infection?

The same question goes for the frequency of double negative T cells.

The number of cells will certainly help in the interpretation of these data. Specifically, it will clear up whether these changes in the frequency of T cells depicts an effect of VHL in the survival, proliferation or recruitment of these populations or because there are altered recruitment of other cells to the lungs that result in altered frequencies of the populations analyzed, or both.

The RAG^{-/-} transfer is nice to show the key role of CD4 T cells in the phenotype of these mice. It would have been interesting to have RAG^{-/-} mice transferred with WT or VHL cKO CD4 T cells that are left uninfected to see the take of the transferred cells. As naïve T cells will expand in an empty mouse, in an Ag-independently way, it would also help clear up whether the effect seen in VHL cKO cells is indeed dependent on TCR signaling.

Fig2 is clear, although the numbers of cells would be interesting as well but not as important as before. The Annexin V staining appears to be in total CD4 T cells. How about Ag-specific cells (may not be possible)? Would be interesting to see if Ag-specific cells are the ones dying. In any case, it would be interesting to see if Annexin V stains both naïve and Ag-experienced T cells (CD44⁺ at least).

Fig3 is very confusing as it is not very clear the message the authors are trying to convey. Are they suggesting the effects of VHL-deficiency is more pronounced in specific populations of T cells? the increased frequencies of T cells expressing CD44 while maintaining CD62L in the lungs is rather intriguing, as one would expect these cells to be retained in the MLN. As there is an infection going, while this phenotype is characteristic of Tcm cells, these data suggest T cells cannot progress in their differentiation process. Is this the reason for the reduced frequency of Ag-specific cells (fig2), or Ag-specific cells die (Annexin V from fig2)? In line with this question, as there is increased expression of CTLA-4 (which appears to be surface and intracellular expressed?) are the author's suggesting that the reduced frequency of Ag-specific T cells is a consequence of the expression of these inhibitory molecules or because they die (Annexin V fig2)? Or both? The message is not very clear other than the phenotype of the cells is different in VHL cKO mice.

Fig4 OK

Fig5 is an in vitro approach to dissect the mechanism whereby VHL-deficiency influences T cell activation. The data show an increased stabilization of HIF1a (as expected) and a reduction in MYC (in line with the RNAseq data); consequently, T cells display deficient proliferation. These results are also in line with the in vivo data showing that T cells do not differentiate properly (maintain CD62L, etc...), likely because they cannot proliferate and end up dying. However, the dramatic effect of VHL cKO in the MLN (fig1) suggest that the effect may go beyond Ag-specific cells. As such, these in vitro experiments should be carried out with PMA/IONO in parallel with aCD3/aCD28 to clarify if it is indeed TCR dependent or any type of inflammatory trigger (including cytokines) can induce the expression of HIF-1a and impact the response of T cells.

The Issues of Fig5 above are also true for fig6. In this figure is not clear whether T cell cultures have exogenous IL-2 (the discussion clarifies this). Regardless, as there is a significant reduction in IL-2, the experiment adding exogenous IL-2 should be depicted. The final panels of this figure should discriminate CD45.1 and CD45.2 cells in the mixed adoptive transfer (despite the fact that it is likely that the cells come from the WT genotype). One important point here is that these mice were not infected and yet VHL cKO cells do not expand. Again, these data go against the TCR-triggering requirements that the authors suggest...

Fig7 OK

Not clear what the message is with Fig8. The analysis is done at 3 wks post immunization, which is still in the effector phase of the response (maybe towards the end?). BCG CFU's would have been nice to clarify this. As such, it is not surprising that the data are similar to the data shown for Mtb infection. It would be interesting to show whether there is an impact in the long-term, including in the establishment of a memory response followed by an Mtb challenge to determine the protective role of the recall response.

Fig9 OK

Just a few points for the discussion: in the introduction, the importance of vaccination was brought up, but it is not clear on how these new data can be applied to vaccination. In addition, previous data suggest that the absence VHL improves clearance of tumors and viral infection, but this is not the case in TB. These two points may merit discussing in the light of these data.

Reviewer #2 (Remarks to the Author):

A manuscript on the role and regulation of Hypoxia inducing factors on susceptibility to experimental tuberculosis in mice is presented. The major findings are

1. Deletion of Vhl increases the CFU and decreases the total, and antigen specific numbers and phenotype of both CD4 and CD8 T cells in the organs of infected mice
2. By contrast deletion of HIF-1 alpha has no effect on CFU counts and a double KO of HIF and Vhl also has no phenotype. The authors interpret this to indicate HIF-1 is mediating the susceptibility to M. tuberculosis infection. The immune effects of BCG vaccination are also unaffected in the double KO mouse.
3. Adoptive transfer on WT but not Vhl KO cells can partially rescue Rag -/- mice
4. There is an increase in HIF-inducible transcripts in the lungs of Vhl KO mice
5. Further experiments show that Vhl is required for TcR activation of CD4 cells and promotes their proliferation

The manuscript presents a lot of data on both M. tuberculosis, PMA and Anti-CD28 stimulated cells both ex vivo and in vivo. These experiments tend to towards the conclusions above the discussion places the findings in context. Overall, I could not really fault the experiments or the interpretation. As the granuloma is known to be markedly hypoxic these findings are interesting, novel and potentially important. Future experiments in human cells and analyses of infected tissues would be informative.

POINT-TO-POINT RESPONSES TO REVIEWER COMMENTS

Reviewer #1 (Remarks to the Author):

This study uses a mouse model of TB infection with a conditional T cell KO in VHL or Hif-1 (or both) to examine the effect of HIFs the immune response to Mtb infection. There are a series of systematic experiments that reveal a negative effect of VHL conditional KO in T cells in the control of Mtb infection, which appears to be mediated by the stabilization of Hif-1a. This study follows up several other studies, particularly on the role of HIF-1a, which are not conclusive and somewhat contradictory, thus providing additional information related to the complex role of HIFs in TB. The manuscript is well written and the story is easy to follow; the experimental design and data analysis are robust.

The first few panels of fig 1 are very clear and show the increased susceptibility of VHL conditional KO mice to Mtb infection, with elevated bacterial burdens decreased survival and weight and increased lung pathology. On the other hand, Hif-1a cKO display the same bacterial burdens as control mice. From this point on, including the data from the next few figures, the message is not very clear, as it gets lost in so many panels showing mostly frequencies of different populations of T cells. Only in very few occasions, the number of cells is depicted. Unarguably, the frequency of total CD4 and CD8 T cells is relevant here (certainly before Mtb infection, to show that VHL or HIF-1a cKO mice have normal frequencies of T cells), but the numbers of cells are equally relevant to understand what do these reduced frequencies mean. For instance, the authors show a dramatic reduction of CD4 T cells in the lungs of VHL cKO mice after Mtb infection but is the number of these cells reduced as dramatically? If so, is this reduction in activated T cells specifically, or are both naïve and activated T cells affected equally? Based on these data and the next figure one would expect the Ag-specific population to be affected the most (if not uniquely). However, the authors show a 50% reduction in CD4 T cells in the MLN, and it is very unlikely that 50% of CD4 T cells in the MLN are Mtb-specific. This is important to clarify as the main message of the paper that suggest that the effects of VHL cKO is dependent on TCR activation (which is not clear, see below) but if this is indeed the case how can one explain the dramatic reduction of CD4 T cells in the MLN, at 8 wks post-infection?

We found that the numbers of CD3 T cells in the MLN of WT and *Vhl* cKO mice before or after infection with *M. tuberculosis* were similar. The numbers of CD4 T cells were reduced in the MLN from *Vhl* cKO mice compared to those in WT controls after (but not before) *M. tuberculosis* infection (Supplementary Figure 1j, k). An increased numbers of T cells (and of CD4 T cells) was recorded in the MLN from WT and mutant mice after *M. tuberculosis* infection, although Ki-67+ and tetramer + cells were lower in *M. tuberculosis*-infected *Vhl* cKO mice.

The current dogma indicates that T cells present in the lymph nodes can be stimulated by antigen to divide, produce effector cytokines, and migrate to peripheral tissues. By contrast, activated T cells that had migrated into non-lymphoid tissues (such as the lung) produce substantial effector cytokines upon antigen challenge, but are unable to divide or migrate back to the lymph nodes. However, we and others have previously reported that mycobacteria-specific CD4 and CD8 T cells preferentially accumulated in the lung as compared to the MLN at different time points

after *M. tuberculosis* infection where they proliferate^{1, 2}. Chemical inhibition of recirculation indicated that resident memory T cells were generated in the lung after *M. tuberculosis* infection. Thus, as indicated by the reviewer, specific T cells do not constitute the majority of MLN cells and probably do not account for the reduction observed. The final number of T cells result probably from differences in proliferation to antigen, the ability to undergo homeostatic growth, lymph node retention and death.

The same question goes for the frequency of double negative T cells. The number of cells will certainly help in the interpretation of these data. Specifically, it will clear up whether these changes in the frequency of T cells depicts an effect of VHL in the survival, proliferation or recruitment of these populations or because there are altered recruitment of other cells to the lungs that result in altered frequencies of the populations analyzed, or both.

The frequency and numbers of the double negative *Vhl cKO* T cells are now provided in supplementary figures 1e, f. The frequency of double negative cells in lungs from *M. tuberculosis* infected mice increased from 5% in the WT to more than 50 % in *Vhl cKO* mice, while the number of DN cells increased 30 times (Supplementary figure 1f). The double negative cells consisted of 50% $\gamma\delta$ T cells in WT and more than 95% $\gamma\delta$ T cells in *vhl cKO*.

The *cd4 cre* transgene is expressed at later stages of thymic development than the β -selection when $\gamma\delta$ T cells become committed. Thus, $\gamma\delta$ T cells will not display *Vhl* loss in the *Vhl^{fl/fl} cd4 cre* model. The increased $\gamma\delta$ T cell levels in lungs from *M. tuberculosis*-infected *Vhl cKO* mice is thus a probable consequence of the different dynamics of the antigen (bacteria) or the inflammatory responses in these mice.

The RAG^{-/-} transfer is nice to show the key role of CD4 T cells in the phenotype of these mice. It would have been interesting to have RAG^{-/-} mice transferred with WT or VHL cKO CD4 T cells that are left uninfected to see the take of the transferred cells. As naïve T cells will expand in an empty mouse, in an Ag-independently way, it would also help clear up whether the effect seen in VHL cKO cells is indeed dependent on TCR signaling.

The experiment requested by the reviewer is shown in Figure 6o, p. Our results indicate that WT but not *Vhl cKO* CD4T cells are able to reconstitute blood and spleens from uninfected *rag2^{-/-}* mice. Using mice deficient in MHC class I or II molecules, it has been shown that homeostatic proliferation of CD4 and CD8 T cells requires the contact with self-MHC class II and I molecules, respectively. Studies from several laboratories strongly suggest that homeostatic proliferation is driven by low-affinity interactions with self-MHC molecules loaded with self-peptides probably the peptides that were involved in the positive selection in the thymus^{3, 4, 5, 6}. Thus, a defective homeostatic proliferation of *Vhl cKO* does not necessarily reflect a TCR-independent process, although antigen has not been inoculated. We have discussed this point further in the discussion section of our revised manuscript.

Fig2 is clear, although the numbers of cells would be interesting as well but not as important as before. The Annexin V staining appears to be in total CD4 T cells. How about Ag-specific cells (may not be possible)? Would be interesting to see if Ag-

specific cells are the ones dying. In any case, it would be interesting to see if Annexin V stains both naïve and Ag-experienced T cells (CD44+ at least).

We did not have the technical possibility of measuring Annexin V labelling in tetramer binding T cells in our BSL3 flow-cytometer. In figure 6n the fraction of Annexin V+ expressing cells after in vitro stimulation is gated on CD44+ CD4 T cells, information that was added in the figure caption.

Fig3 is very confusing as it is not very clear the message the authors are trying to convey. Are they suggesting the effects of VHL-deficiency is more pronounced in specific populations of T cells? the increased frequencies of T cells expressing CD44 while maintaining CD62L in the lungs is rather intriguing, as one would expect these cells to be retained in the MLN. As there is an infection going, while this phenotype is characteristic of T_{CM} cells, these data suggest T cells cannot progress in their differentiation process.

Is this the reason for the reduced frequency of Ag-specific cells (fig2), or Ag-specific cells die (Annexin V from fig2)?

We have tried to improve the clarity of figure 3, by removing several panels to the supplementary figure s2 and figure s7 and group panels that address features of similar biological significance. The main messages are the differences in the frequencies of memory T cell populations between WT and *Vhl cKO* T cells being the most sticking one the presence of a large CD8 T cell population expressing CD44 and CD62L in mice with *Vhl* loss, which is also observed in uninfected animals. The other feature is the increased expression of inhibitory receptors and lower levels of markers of full or partial differentiation in *Vhl cKO* T cells.

Increased levels of CD62L are mainly observed in CD8 T cells. These cells are also increased in the uninfected/ untreated *Vhl cKO* mice. As suggested by the reviewer we have measured the presence of tetramer + cells within T_{CM} and T_{EM} populations in *Vhl cKO* mice. We observed a higher frequencies of TB10.4 binding CD8 T cells in T_{EM} than within T_{CM} cells in both WT and *Vhl cKO* mice. Higher frequencies of TB10.4 tetramer binding CD8 T cells were measured in WT than in the *Vhl cKO* T_{EM}, while the percentage of TB10.4-specific CD8 T cells in WT and mutant T_{CM} cells was similar (Figure 3d).

WT — *Vhl cKO* —

Figure 3d. The mean frequencies of tetramer TB10.4-binding CD8 T_{CM} and T_{EM} cells (d) in the lungs from *Vhl* cKO and WT mice 8 weeks after infection with *M. tuberculosis* are depicted. Differences in frequencies are significant at *** $p \leq 0.001$ unpaired t-test with Welch correction for multiple comparisons.

Apart from canonical, antigen-experienced memory T cells, some CD8T cells may exhibit a memory phenotype without overt immunization or infection. These cells, unlike true memory T cells that develop in response to foreign antigen, express only low levels of CD49d and are termed virtual memory T (T_{VM}) cells^{7,8,9}. Spleen and lung CD44+CD62L+ CD8 T cells in non-infected *Vhl* cKO mice expressed low levels of CD49d suggesting that these are virtual memory T cells (figure 8l and supplementary figure 7i). The percentage of CD49d+ within CD44+CD62L+ CD8 T cells was higher in lungs from WT than in those from *vhl* cKO BCG-immunized mice (Figure 8l). In normal mice virtual memory T cells have been shown to comprise 5-20% CD8 T cells, and have been shown to preferentially differentiate into CD8T_{CM}^{10,11}

Figure 8l. The mean percentage of lung CD49d+ CD8 T_{CM} before and after BCG immunization of WT and *Vhl* cKO mice are shown. Differences between groups are significant at *** $p \leq 0.001$, unpaired t-test with correction.

Supplementary fig. 7i. Representative histogram of the expression of CD49d in spleen cells from *vhl* cKO and WT non-immunized mice is shown.

In line with this question, as there is increased expression of CTLA-4 (which appears to be surface and intracellular expressed?) are the author's suggesting that the reduced frequency of Ag-specific T cells is a consequence of the expression of these inhibitory molecules or because they die (Annexin V fig2)? Or both? The message is not very clear other than the phenotype of the cells is different in VHL cKO mice.

We hypothesize that the low number of specific *Vhl* cKO CD4T cells *in vivo* is due to a defect in their expansion and that the responses *in vivo* reflect the observations *in vitro*, as also shown by the defective reconstitution and expansion of donor T cells after infection. As indicated by the reviewer, we observe the presence of CD44+ *Vhl* cKO T cells with high levels of CTLA-4, PD-1, that might impair responses to antigen, and their ability to fully differentiate to effectors as indicated by the low KLRG1 and CX3CR1 expression levels. These two molecules are markers of full or partial T cell differentiation. Thus, T cells that still develop in *Vhl* cKO mice undergo an aberrant differentiation program. We have clarified this interpretation in the discussion of our revised manuscript.

The levels of CTLA4 expression in CD4 or CD8 T cells from uninfected WT and *Vhl* cKO mice were found to be negligible as compared to those from *M. tuberculosis*-infected mice, and this information was added to the revised manuscript (Figure 3n, o). We have stained intracellular CTLA-4 levels as indicated in the methods section. Thus we labelled both intra and extracellular CTLA-4 molecules.

Figure 3n, o. The mean fractions of CTLA-4+ CD4 (n) and CD8 (o) T cells in the lungs of mice before and 8 weeks after infection with *M. tuberculosis* are depicted. Differences between groups are significant at *** $p \leq 0.001$, unpaired *t*-test with correction.

While PD-1 levels were not increased in the BCG-immunized mice, the frequencies of CTLA-4+ CD4 and CD8 T cells from lungs of BCG-immunized *Vhl* cKO mice

were higher than those measured in WT controls, which is now included in the revised version of our manuscript (Fig. 8q).

Fig4 OK

Fig5 is an in vitro approach to dissect the mechanism whereby VHL-deficiency influences T cell activation. The data show an increased stabilization of HIF1a (as expected) and a reduction in MYC (in line with the RNAseq data); consequently, T cells display deficient proliferation. These results are also in line with the in vivo data showing that T cells do not differentiate properly (maintain CD62L, etc...), likely because they cannot proliferate and end up dying. However, the dramatic effect of VHL cKO in the MLN (fig1) suggest that the effect may go beyond Ag-specific cells. As such, these in vitro experiments should be carried out with PMA/IONO in parallel with aCD3/aCD28 to clarify if it is indeed TCR dependent or any type of inflammatory trigger (including cytokines) can induce the expression of HIF-1a and impact the response of T cells.

We have compared the responses to anti-CD3/ CD28 of WT, *Vhl cKO* and *Vhl Hif1a dcKO* CD4 T cells in figure 9. We observed that while the frequency of WT, *Vhl cKO* and *Vhl Hif1a dcKO* CD4 T cells expressing phospho-rpS6 after anti-CD3/ CD28 stimulation was similar, the levels of expression were reduced in *Vhl cKO* as compared to WT and *Vhl Hif1a dcKO* CD4 T cells (Figure 9l, m and Figure S8k). Instead, the levels of phospho-rpS6 expression in WT and mutant CD4 T cells after PMA/ I stimulation were similar (Figure 9n), suggesting that PMA/ I-activation bypasses the HIF-1-mediated inhibition of a TCR proximal signaling. In line with this, the frequency of IFN-g-secreting *Vhl cKO* CD4 T cells after anti-CD3/CD28 stimulation was lower than those from WT or *Vhl Hif1a dcKO* CD4 T cells, while the frequency of IFN-g secreting *Vhl cKO*, *Vhl Hif1a dcKO* and WT CD4 T cells stimulated with PMA/ I was similar (Figure 9p), also suggesting that T cell activation in response to PMA/I is not impaired in *Vhl cKO* CD4 T cells.

Following the suggestion of the reviewer, we have compared the expression of CD44, CD69 and CD25 on WT and *Vhl cKO* CD4T cells after PMA/ Ionomycin stimulation. We observed similar expression levels of CD44, CD69 and CD25 in *Vhl cKO* and WT CD4 T cells after PMA/ I stimulation (Supplementary figure 9l-n). Further, we also

showed that the levels of CD44, CD69, CD25 as well as the proliferative responses were diminished in *Vhl* cKO CD4 T as compared to WT T cells stimulated with the bacterial Staphylococcal enterotoxin B superantigen (that activates T cells by binding to certain TCR V β s) (Supplementary figure s9a-k). These data confirm that VHL allows proper TCR signaling.

Supplementary figure 9. VHL expression allows TCR-mediated activation of CD4 T cells

(a-i) The percentage of CD44, CD69 and CD25 positive cells (a, d, g) and the level of expression of these markers (b, e, h) were determined in *Vhl* cKO and WT CD4 T cells 72 h after incubation with either anti-CD3/CD28, SEB or control medium. (c, f, i) Representative histograms of CD44, CD69 and CD25 expression on WT and *Vhl* cKO CD4 T cells 72 h after SEB-stimulation are shown. (j, k) Representative profiles (k) of CFSE-labelled *Vhl* cKO and WT CD4 T cells 3 days after stimulation with anti-CD3/CD28 or SEB and the mean division index \pm SEM (j) are shown. (l-n) The percentage of CD44 (l), CD69 (m) and CD25 (n) expressing WT and *Vhl* cKO CD4 T cells were determined 24h after PMA/ Ionomycin stimulation. Differences between groups are significant at $*p \leq 0.05$, $**p \leq 0.01$ and $***p \leq 0.001$ unpaired *t*-test with Welch correction.

We also include data showing that the level of CD4 in *Vhl* cKO T cells is low (Supplementary Fig. 4e, f), complementing our results indicating reduced levels of CD3 ϵ and TCR- β density in non-stimulated *Vhl* cKO CD4 T cells (Figure 5j-m).

Supplementary figure 4e, f. Representative histograms (e) and the mean MFI \pm SEM (f) of the CD4 expression in *Vhl* cKO and WT CD4 T cells are shown. Differences are significant at $*p \leq 0.05$ unpaired Welch *t* test.

The Issues of Fig5 above are also true for fig6. In this figure is not clear whether T cell cultures have exogenous IL-2 (the discussion clarifies this). Regardless, as there is a significant reduction in IL-2, the experiment adding exogenous IL-2 should be depicted. The final panels of this figure should discriminate CD45.1 and CD45.2 cells in the mixed adoptive transfer (despite the fact that it is likely that the cells come from the WT genotype). One important point here is that these mice were not infected and yet VHL cKO cells do not expand. Again, these data go against the TCR-triggering requirements that the authors suggest...

Our data was generated in the absence of exogenous IL-2. As suggested by the reviewer we have included experiments adding 20 ng/ ml rec IL-2 into the anti-CD3/CD28-stimulated WT and *Vhl* cKO CD4 T cell cultures. We determined that the addition of exogenous IL-2 did neither restore the proliferation, the CD44 expression nor resulted in increased cell size of TCR-stimulated *Vhl* cKO CD4 T cells (Supplementary Fig. 5i-k).

Supplementary fig.5i-k. WT and *Vhl* cKO CD4 T cells were stimulated with anti-CD3/CD28 in presence or absence of 20 ng/ ml IL-2. The mean percentage of blast cells (i), the mean division index calculated by CFSE dilution (j) and the percentage of cells expressing CD44 (k) were determined 3 days after TCR-stimulation.

We have also included the CD45.1 and CD45.2 CD4T cell discrimination in the mixed adoptive transfer when measuring homeostatic proliferation. While the inoculated cell populations contained even numbers of CD45.1 and CD45.2 populations, CD4 T cells from the spleen or blood from transferred animals were CD45.1+ WT genotype indicating that the WT CD4T cells do not complement the proliferative defect of *Vhl* cKO CD4 T cells (Supplementary figure 5n).

Supplementary fig. 5n. The representative dot plots of CD45.1 and CD45.2 in CD4 T cells before being co-transferred or in the blood and spleens of mice 5 weeks after co-transfer (n) are shown.

Please refer to the answer to figure 1 with regards to the interpretation of homeostatic proliferation as a T cell receptor signaling-dependent event.

Fig7 OK

Not clear what the message is with Fig8. The analysis is done at 3 wks post immunization, which is still in the effector phase of the response (maybe towards the end?). BCG CFU's would have been nice to clarify this. As such, it is not surprising that the data are similar to the data shown for Mtb infection. It would be interesting to show whether there is an impact in the long-term, including in the establishment of a memory response followed by an Mtb challenge to determine the protective role of the recall response.

We agreed with the comment of the reviewer. Previous reports have shown that the BCG titers slowly declined, but mycobacteria were still detectable in livers and in spleens nearly 3 months post i.v. inoculation. The bacterial load in the lungs was less than 1% of the load in the spleen or liver ¹². We found that the BCG CFU titers in the spleens and lungs from *Vhl cKO* was higher than those in WT mice, indicating that clearance or dissemination of BCG was also impaired in *Vhl cKO* mice. We have added this information in the revised manuscript (Supplementary figure 7r). In agreement, BCG were also present in organs of WT and *Vhl cKO* mice 6 weeks after immunization, so this time point neither represented a time point where bacteria have been cleared. No overt signs of clinical disease in WT or *Vhl cKO* mice were observed during the experiment. A immunization-challenge study would take several months, so would not be compatible with the time for resubmission. The in vitro experiments indicate defects in T cell responses due to HIF-1 stabilization are also observed when the similar levels of the TCR-stimulus are present.

Supplementary Fig 7r. The log₁₀ BCG CFU in the lungs and spleens of *Vhl cKO* and WT mice at 3 weeks after BCG immunization are depicted. Differences are significant at * $p \leq 0.05$ unpaired Welch *t* test.

Fig9 OK

Just a few points for the discussion: in the introduction, the importance of vaccination was brought up, but it is not clear on how these new data can be applied to vaccination. In addition, previous data suggest that the absence VHL improves clearance of tumors and viral infection, but this is not the case in TB. These two points may merit discussing in the light of these data.

We also agree these points deserve further discussion and will add it to the revised manuscript. We speculate that the differences between our results and those measuring the responses of Vhl deficient T cells to tumors and viral infections in which the absence of Vhl in T cells improves the control may be due to these responses are CD8 mediated. The virtual memory CD8T cells that develop *Vhl cKO* mice may play a role in the improved control of tumors and viruses. Virtual memory CD8T cells have been shown to display low IFN- γ responses but could productively contribute to antigen-specific responses against invading viruses^{13, 14} and other bacteria¹⁵. We have included these points in the discussion of the revised version of our manuscript.

Reviewer #2 (Remarks to the Author):

A manuscript on the role and regulation of Hypoxia inducing factors on susceptibility to experimental tuberculosis in mice is presented. The major findings are

1. Deletion of Vhl increases the CFU and decreases the total, and antigen specific numbers and phenotype of both CD4 and CD8 T cells in the organs of infected mice
2. By contrast deletion of HIF-1 alpha has no effect on CFU counts and a double KO of HIF and Vhl also has no phenotype. The authors interpret this to indicate HIF-1 is mediating the susceptibility to M. tuberculosis infection. The immune effects of BCG vaccination are also unaffected in the double KO mouse.
3. Adoptive transfer on WT but not Vhl KO cells can partially recue Rag -/- mice
4. There is an increase in HIF-inducible transcripts in the lungs of Vhl KO mice
5. Further experiments show that Vhl is required for TcR activation of CD4 cells and promotes their proliferation

The manuscript presents a lot of data on both M. tuberculosis, PMA and Anti-CD28 stimulated cells both ex vivo and in vivo. These experiments tend to towards the conclusions above the discussion places the findings in context. Overall, I could not really fault the experiments or the interpretation. As the granuloma is known to be markedly hypoxic these findings are interesting, novel and potentially important. Future experiments in human cells and analyses of infected tissues would be informative.

References

1. Basile, J.I. *et al.* Mycobacteria-Specific T Cells Are Generated in the Lung During Mucosal BCG Immunization or Infection With Mycobacterium tuberculosis. *Front Immunol* **11**, 566319 (2020).

2. Bull, N.C. *et al.* Enhanced protection conferred by mucosal BCG vaccination associates with presence of antigen-specific lung tissue-resident PD-1(+) KLRG1(-) CD4(+) T cells. *Mucosal Immunol* **12**, 555-564 (2019).
3. Ernst, B., Lee, D.S., Chang, J.M., Sprent, J. & Surh, C.D. The peptide ligands mediating positive selection in the thymus control T cell survival and homeostatic proliferation in the periphery. *Immunity* **11**, 173-181 (1999).
4. Goldrath, A.W. & Bevan, M.J. Low-affinity ligands for the TCR drive proliferation of mature CD8+ T cells in lymphopenic hosts. *Immunity* **11**, 183-190 (1999).
5. Kieper, W.C. & Jameson, S.C. Homeostatic expansion and phenotypic conversion of naive T cells in response to self peptide/MHC ligands. *Proc Natl Acad Sci U S A* **96**, 13306-13311 (1999).
6. Viret, C., Wong, F.S. & Janeway, C.A., Jr. Designing and maintaining the mature TCR repertoire: the continuum of self-peptide:self-MHC complex recognition. *Immunity* **10**, 559-568 (1999).
7. Akue, A.D., Lee, J.Y. & Jameson, S.C. Derivation and maintenance of virtual memory CD8 T cells. *J Immunol* **188**, 2516-2523 (2012).
8. Chiu, B.C., Martin, B.E., Stolberg, V.R. & Chensue, S.W. Cutting edge: Central memory CD8 T cells in aged mice are virtual memory cells. *J Immunol* **191**, 5793-5796 (2013).
9. White, J.T., Cross, E.W. & Kedl, R.M. Antigen-inexperienced memory CD8(+) T cells: where they come from and why we need them. *Nat Rev Immunol* **17**, 391-400 (2017).
10. Haluszczak, C. *et al.* The antigen-specific CD8+ T cell repertoire in unimmunized mice includes memory phenotype cells bearing markers of homeostatic expansion. *J Exp Med* **206**, 435-448 (2009).
11. Lee, J.Y., Hamilton, S.E., Akue, A.D., Hogquist, K.A. & Jameson, S.C. Virtual memory CD8 T cells display unique functional properties. *Proc Natl Acad Sci U S A* **110**, 13498-13503 (2013).
12. Johansen, P. *et al.* Relief from Zmp1-mediated arrest of phagosome maturation is associated with facilitated presentation and enhanced immunogenicity of mycobacterial antigens. *Clin Vaccine Immunol* **18**, 907-913 (2011).
13. Hou, S. *et al.* Virtual memory T cells orchestrate extralymphoid responses conducive to resident memory. *Sci Immunol* **6**, eabg9433 (2021).

14. Rolot, M. *et al.* Helminth-induced IL-4 expands bystander memory CD8(+) T cells for early control of viral infection. *Nat Commun* **9**, 4516 (2018).
15. Drobek, A. *et al.* Strong homeostatic TCR signals induce formation of self-tolerant virtual memory CD8 T cells. *EMBO J* **37** (2018).

REVIEWERS' COMMENTS

Reviewer #1 (Remarks to the Author):

The authors have addressed all the issues raised. The main message of the manuscript is now cleared and, therefore, I have no further comments.